# An Atypical F-Actin Capping Protein Modulates Cytoskeleton Behaviors Crucial for *Trichomonas vaginalis* Colonization

Kai-Hsuan Wang,[a] Jing-Yang Chang,[a] Fu-An Li,[b] Kuan-Yi Wu,[a] Shu-Hao Hsu,[c] Yen-Ju Chen,[a] Tse-Ling Chu,[d] Jessica Lin,[d]
Hong-Ming Hsu[a]

[a]Department of Tropical Medicine and Parasitology, College of Medicine, National Taiwan University, Taipei, Taiwan
[b]The Proteomic Core, Institute of Biomedical Sciences, Academia Sinica, Taipei, Taiwan
[c]Department of Anatomy and Cell Biology, College of Medicine, National Taiwan University, Taipei, Taiwan
[d]Taipei First Girls High School, Taipei, Taiwan

Kai-Hsuan Wang and Jing-Yang Chang contributed equally to this work. Author order was determined in order of seniority.

**ABSTRACT** Cytoadherence and migration are crucial for pathogens to establish colonization in the host. In contrast to a nonadherent isolate of *Trichomonas vaginalis*, an adherent one expresses more actin-related machinery proteins with more active flagellate-amoeboid morphogenesis, amoeba migration, and cytoadherence, activities that were abrogated by an actin assembly blocker. By immunoprecipitation coupled with label-free quantitative proteomics, an F-actin capping protein (*T. vaginalis* F-actin capping protein subunit $\alpha$ [*Tv*FACP$\alpha$]) was identified from the actin-centric interactome. His-*Tv*FACP$\alpha$ was detected at the barbed end of a growing F-actin filament, which inhibited elongation and possessed atypical activity in binding G-actin in *in vitro* assays. *Tv*FACP$\alpha$ partially colocalized with F-actin at the parasite pseudopod protrusion and formed a protein complex with $\alpha$-actin through its C-terminal domain. Meanwhile, *Tv*FACP$\alpha$ overexpression suppressed F-actin polymerization, amoeboid morphogenesis, and cytoadherence in this parasite. Ser2 phosphorylation of *Tv*FACP$\alpha$ enriched in the amoeboid stage of adhered trophozoites was reduced by a casein kinase II (CKII) inhibitor. Site-directed mutagenesis and CKII inhibitor treatment revealed that Ser2 phosphorylation acts as a switching signal to alter *Tv*FACP$\alpha$ actin-binding activity and the consequent actin cytoskeleton behaviors. Through CKII signaling, *Tv*FACP$\alpha$ also controls the conversion of adherent trophozoites from amoeboid migration to the flagellate form with axonemal motility. Together, CKII-dependent Ser2 phosphorylation regulates *Tv*FACP$\alpha$ binding to actin to fine-tune cytoskeleton dynamics and drive crucial behaviors underlying host colonization by *T. vaginalis*.

**IMPORTANCE** Trichomoniasis is one of the most prevalent nonviral sexually transmitted diseases. *T. vaginalis* cytoadherence to urogenital epithelium cells is the first step in the colonization of the host. However, studies on the mechanisms of cytoadherence have focused mainly on the role of adhesion molecules, and their effects are limited when analyzed by loss- or gain-of-function assays. This study proposes an extra pathway in which the actin cytoskeleton mediated by a capping protein $\alpha$-subunit may play roles in parasite morphogenesis, cytoadherence, and motility, which are crucial for colonization. Once the origin of the cytoskeleton dynamics could be manipulated, the consequent activities would be controlled as well. This mechanism may provide new potential therapeutic targets to impair this parasite infection and relieve the increasing impact of drug resistance on clinical and public health.

**KEYWORDS** actin capping protein, actin cytoskeleton, cytoadherence, colonization, *Trichomonas vaginalis*

Address correspondence to Hong-Ming Hsu, hsuhm@ntu.edu.tw.

The authors declare no conflict of interest.

*T*richomonas vaginalis is a pathogenic protist causing trichomoniasis, which is one of the most prevalent nonviral sexually transmitted diseases, with approximately 180 million new infections worldwide annually (1).

A successful pathogenic infection includes cytoadherence to establish colonization, followed by migration for population spread. Numerous studies on *Trichomonas vaginalis* have focused on the cytoadherence mechanism in adhesion molecules like cadherin (2), rhomboid protease (3), legumain protease (4), BAP proteins (5), *T. vaginalis* AD1 (*Tv*AD1) protein (6), and surface-expressed hydrogenosomal proteins (7–10). However, the effects of these reputed adhesins in cytoadherence are limited when analyzed by gain- or loss-of-function assays (2–10). Thus, we postulated that the cytoadherence of *T. vaginalis* might be regulated by pathways other than adhesion molecules. In mammalian adhesion cells, transmembrane integrins link peripheral focal protein complexes underneath the cell membrane for focal adhesion, which is the site that connects the extracellular matrix to transmit traction forces required for cell migration and activates downstream signaling followed by local cytoskeleton reorganization (11–13). A few studies have used ligand competition or antibody neutralization to demonstrate the involvement of integrin-like molecules in the cytoadherence of *T. vaginalis* (14–16). Recently, the adherence of clinical *T. vaginalis* isolates to a plastic surface or host cells was shown to be influenced by an actin polymerization blocker (17, 18), implying that the actin cytoskeleton might coordinate cytoadherence in *T. vaginalis*, but the regulatory mechanism was unknown.

Furthermore, the flagellate-amoeboid transition immediately after contact with a solid surface or human vagina epithelium cells (hVECs) is another striking feature of adherent isolates of *T. vaginalis* (18, 19). Upon morphological transformation, the free-swimming flagellar trophozoite converts to an adherent trophozoite that crawls over a solid surface by pseudopod-like protrusions, referred to as amoeboid migration. A similar flagellate-amoeboid transition was observed in the pathogenic amoeba *Naegleria fowleri*. This free-living trophozoite builds lamellipodium-like protrusions for phagocytosis and migration driven by actin cytoskeleton machines (20), in which actin expression is correlated with its virulence (21).

The actin cytoskeleton is a complex network of actin filaments and actin-associated proteins that shape cell morphology, drive cellular locomotion, and confer cell adhesion (22–24). The globular actin monomer (G-actin) polymerizes into filamentous actin polymers (F-actin), which are further organized into bundles or branched into three-dimensional networks for complicated cytoskeleton activities. In the polarized F-actin filament, growth initiates from the assembly of the Arp2/3 nucleation complex (25), and G-actin is then continuously added at the fast-growing barbed end or dissociated from the pointed end (26). The cellular actin cytoskeleton dynamics are tightly modulated by a variety of accessory effectors for actin polymerization, depolymerization, branching, and reorganization (27). In higher eukaryotes, F-actin capping protein (CP) is heterodimerized from the $\alpha$-subunit (CP$\alpha$) and $\beta$- subunit (CP$\beta$) to form a mushroom-shaped structure capping the fast-growing barbed end of F-actin to block off G-actin access and subsequent polymerization. The C-terminal regions of CP$\alpha$ and CP$\beta$ form as two tentacles to bind actin (28–30). A set of regulatory proteins binds to the barbed end of F-actin to prevent the binding of CP, or several proteins directly bind CP to spatially guide subcellular localization or allosterically alter actin capping activity for instant cytoskeleton regulation (31, 32).

Posttranslational modifications like phosphorylation and acetylation on the interacting interface within the C-terminal tentacle of CP$\beta$ alter the actin-binding dynamics (33). Human CP$\alpha$ forms a protein complex with casein kinase II (CKII)-interacting protein 1 (CKIP-1) and CKII. CKII phosphorylates Ser9 of CP$\alpha$ and coordinates with CKIP-1 to inhibit capping activity, but Ser9 phosphorylation seems to be independent of CP$\alpha$ capping activity (34, 35). The capacity of CP to bind actin filaments might be regulated in a spatial or allosteric manner to fine-tune the actin assembly dynamics in cells.

The mechanisms of actin cytoskeleton regulation in *T. vaginalis* have not been fully

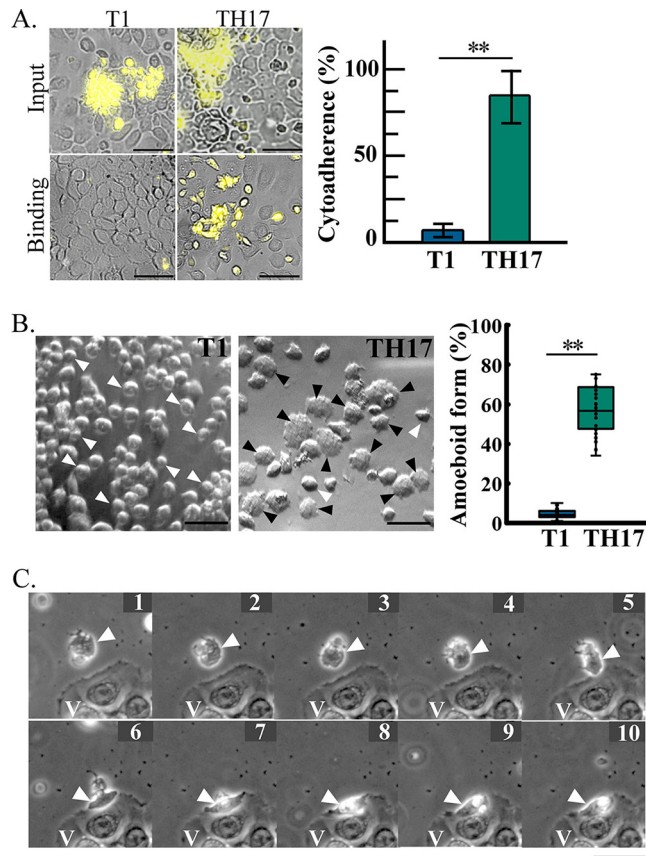

**FIG 1** Differential cytoadherence, morphogenesis, and migration modes of *T. vaginalis*. A variety of behaviors were observed in the nonadherent (T1) and adherent (TH17) isolates. (A) CFSE-preloaded trophozoites were cultured with hVECs and then fixed at 1 h postinfection. The cytoadherence capacity was evaluated by the ratio of binding to input trophozoites, as shown in the bar graph. Bars, 100 $\mu$m. (B) The ratio of T1 or TH17 trophozoites in the amoeboid form was measured in ~600 trophozoites from 12 random microscopic fields, as shown in the box-and-whisker plot. The black and white arrowheads indicate representative amoeboid and flagellate trophozoites, respectively. Bars, 20 $\mu$m. (C) A TH17 trophozoite (white arrowheads) was cocultured with hVECs (V). The dynamics of amoeboid migration and morphogenesis were recorded by time-lapse imaging at 1 frame per 15 s over 10 min. The observation time point (minutes) is indicated at the top right of each image. Bars, 20 $\mu$m. All micrographs were captured in a single z-slice. The assays were performed with three biological repeats ($n = 3$). Data in the bar graphs are presented as means $\pm$ standard deviations (SD). Statistical significance for each group of data was analyzed by Student's $t$ test, as indicated ($n = 3$) (**, $P < 0.01$).

elucidated. The *T. vaginalis* fimbrin 1 protein (*Tv*Fim1) has been identified *in vitro* to accelerate actin assembly and *in vivo* to colocalize with F-actin at the cell membrane periphery in the pseudopod-like structures of *T. vaginalis* upon phagocytosis or migration (36). In this study, a putative F-actin capping protein subunit $\alpha$ (*Tv*FACP$\alpha$) was identified from an $\alpha$-actin-associated protein complex in *T. vaginalis*.

## RESULTS

**Differential morphogenesis, cytoadherence, and motility of *T. vaginalis*.** The differential host-parasite interaction between nonadherent T1 and adherent TH17 isolates was evaluated by cytoadherence, morphogenesis, and motility. Carboxyfluorescein succinimidyl ester (CFSE)-labeled trophozoites were cocultured with an hVEC monolayer at a multiplicity of infection (MOI) of 2:1. Sixty minutes after infection, ~80% of TH17 but few T1 trophozoites bound to the hVEC monolayer (Fig. 1A). Most T1 trophozoites maintained an oval-shaped flagellate form, but ~60% of TH17 trophozoites transformed into a flat disk or irregular amoeboid form and tightly adhered to the slide surface (Fig. 1B). To observe the dynamics of the host-parasite interaction, the trophozoites cocultured with hVECs were monitored by time-lapse imaging (Fig. 1C; see also Videos S1 and S2 in the supplemental

material), showing that nonadherent T1 trophozoites maintained a flagellate form and swam by flagellar locomotion, only randomly coming into contact with the hVECs. In contrast, adherent TH17 trophozoites rapidly transformed into an amoeboid form within 10 min of contact with the glass slide and crawled toward hVECs via pseudopod-like protrusions, referred to as amoeboid migration. The adherent isolate displayed more active cytoadherence and amoeboid morphogenesis and migration.

**Differential expression of actin-related proteins in *T. vaginalis*.** The cytoadherence and migration of *T. vaginalis* are correlated with the actin cytoskeleton (17, 36); therefore, the expression of $\alpha$-actin and $\alpha$-actinin, the major component and actin bundle linker protein in the cytoskeleton, respectively, were investigated. The expression levels of $\alpha$-actin and $\alpha$-actinin were higher in the adherent TH17 and T016 isolates than in the nonadherent T1 isolate and especially a fresh adherent isolate from a clinical vaginitis patient (Fig. 2A). In contrast to the T1, T016, and TH17 experimental long-term-cultured strains, this undomesticated isolate (FC) cultured short term for weeks may have intrinsic virulence, which is correlated with its $\alpha$-actin and $\alpha$-actinin levels.

The immunostaining of $\alpha$-actin was more intense in TH17 than in T1 trophozoites and was detected in the cytoplasm in tiny punctate or short bundles of the flagellate TH17 isolate but in the cytoplasm with sporadic clumps underneath the plasma membrane of the amoeboid TH17 isolate (Fig. 2B). However, the expression levels of $\alpha$-actin and $\alpha$-actinin were similar between the two forms of TH17 trophozoites (Fig. 2C). The phalloidin-binding sites are conserved in $\alpha$-actin of *T. vaginalis* (Fig. S1) (37). F-actin was doubly stained by tetramethylrhodamine isocyanate (TRITC)-conjugated phalloidin and anti-$\alpha$-actin antibody (Fig. 2D), showing prominent F-actin and $\alpha$-actin signals concentrated in the juxtanuclear region, referred to as the perinuclear actin cap (38), with intense staining underneath the cell membrane of the leading edge in protrusive pseudopods and less intense staining in the cytoplasm. The signal colocalization of $\alpha$-actin and phalloidin had a Pearson correlation coefficient value of 0.95 (Fig. 2Da and b). The evaluation of F-actin assembly by fractionation and Western blotting revealed F-actin ratios of $\sim$46.6% $\pm$ 6.1% for the adherent isolate and $\sim$17% $\pm$ 8.2% for the nonadherent one (Fig. 2E), similar to $\alpha$-actinin, indicating that F-actin polymerization is more active in the adherent isolate.

**Actin-based morphogenesis, migration, and cytoadherence in *T. vaginalis*.** Latrunculin B (LatB)-binding sites are conserved in *T. vaginalis* $\alpha$-actin (Fig. S1). TH17 trophozoites were treated with LatB to study their cytoskeleton activities. LatB treatment little changed parasite viability (Fig. S2A) but reduced the F-actin ratio (Fig. 3A) and morphogenesis (Fig. 3B) in the parasite. Also, LatB decreased the wound closure rate (Fig. 3C) and cytoadherence at 60 min postinfection (Fig. 3D), showing that F-actin disorder retarded parasite morphogenesis, amoeboid migration, and cytoadherence. The more intense phalloidin signal distributed around the cell membrane in the dimethyl sulfoxide (DMSO) control was condensed into numerous puncta in the LatB-treated parasites (Fig. S2B), suggesting that LatB modifies the assembly or distribution of parasite F-actin. To rule out effects from the reputed adhesion molecules (2, 7, 10), the expression of AP65 and Pyruvate:ferredoxin oxidoreductase (PFO) (Fig. 3E) and their surface distributions (Fig. 3F) were analyzed, showing that the adhesion molecules were unchanged in the trophozoites with or without LatB treatment. Immunofluorescence assay (IFA) permeabilization conditions proved the specificities of the AP65 and PFO surface signals (7, 10). Also, the surface hemagglutinin (HA)-tagged cadherin-like protein (CLP) was not affected by LatB treatment (Fig. 3G). Taken together, actin polymerization is positively associated with the parasite morphological transition, amoeboid migration, and cytoadherence, independent of adhesion molecules.

**TvFACP$\alpha$ as an $\alpha$-actin effector.** We attempted to identify the regulatory proteins in the $\alpha$-actin-associated complexes. HA-*Tv*Actin was immunoprecipitated from transgenic TH17 trophozoites for mass spectrometry analysis (Fig. S3A), identifying 41 $\alpha$-actin-associated proteins with an exponentially modified protein abundance index (emPAI) score of >0.25 or that were specific in the immunoprecipitant of HA-*Tv*Actin (Table 1). These proteins were classified into multiple cellular pathways, including cytoskeleton proteins (22%),

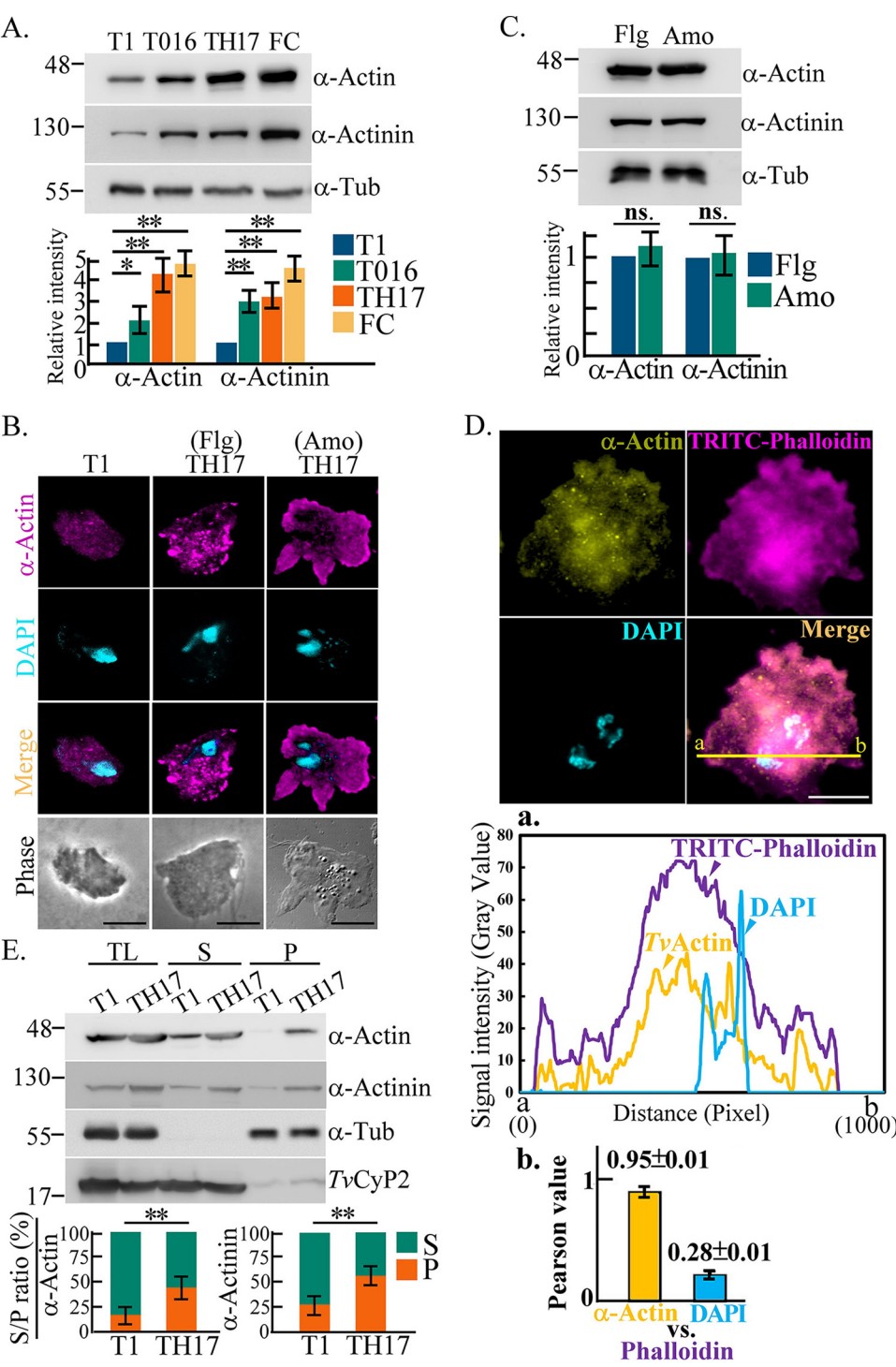

**FIG 2** Differential expression of actin-based machinery proteins in *T. vaginalis*. (A) The total lysates from the T1, T016, and TH17 isolates and a fresh clinical isolate (FC) were subjected to Western blotting. (B and C) TH17 flagellate (Flg) trophozoites suspended in the medium or amoeboid (Amo) trophozoites adhered to the glass surface were sampled for IFAs (B) or Western blotting (C). Bars, 5 $\mu$m (B). (D) TH17 trophozoites cultured on a glass slide and fixed for IFA double staining with anti-$\alpha$-actin antibody and TRITC-conjugated phalloidin. Bars, 2 $\mu$m. (a) Signal colocalization was evaluated by plot profile analysis to show the signal intensity distribution on the yellow line between the a and b sites, as shown in the middle diagram. (b) The colocalization of phalloidin with $\alpha$-actin or DAPI was evaluated by Pearson's correlation coefficient. The micrographs in panels B and D were captured in a single z-slice. (E) The protein lysates of actin fractionation from T1 and TH17 trophozoites were examined by Western blotting. The signal ratios of the indicated proteins in the supernatant (S) and pellet (P) fractions were analyzed, as shown in the bar graphs. All experiments were performed with three biological repeats ($n = 3$). Data in the bar graphs are presented as means $\pm$ SD. Statistical significance for each group of data was measured by Student's *t* test, as indicated ($n = 3$) (**, $P < 0.01$; *, $P < 0.05$; ns, no significance).

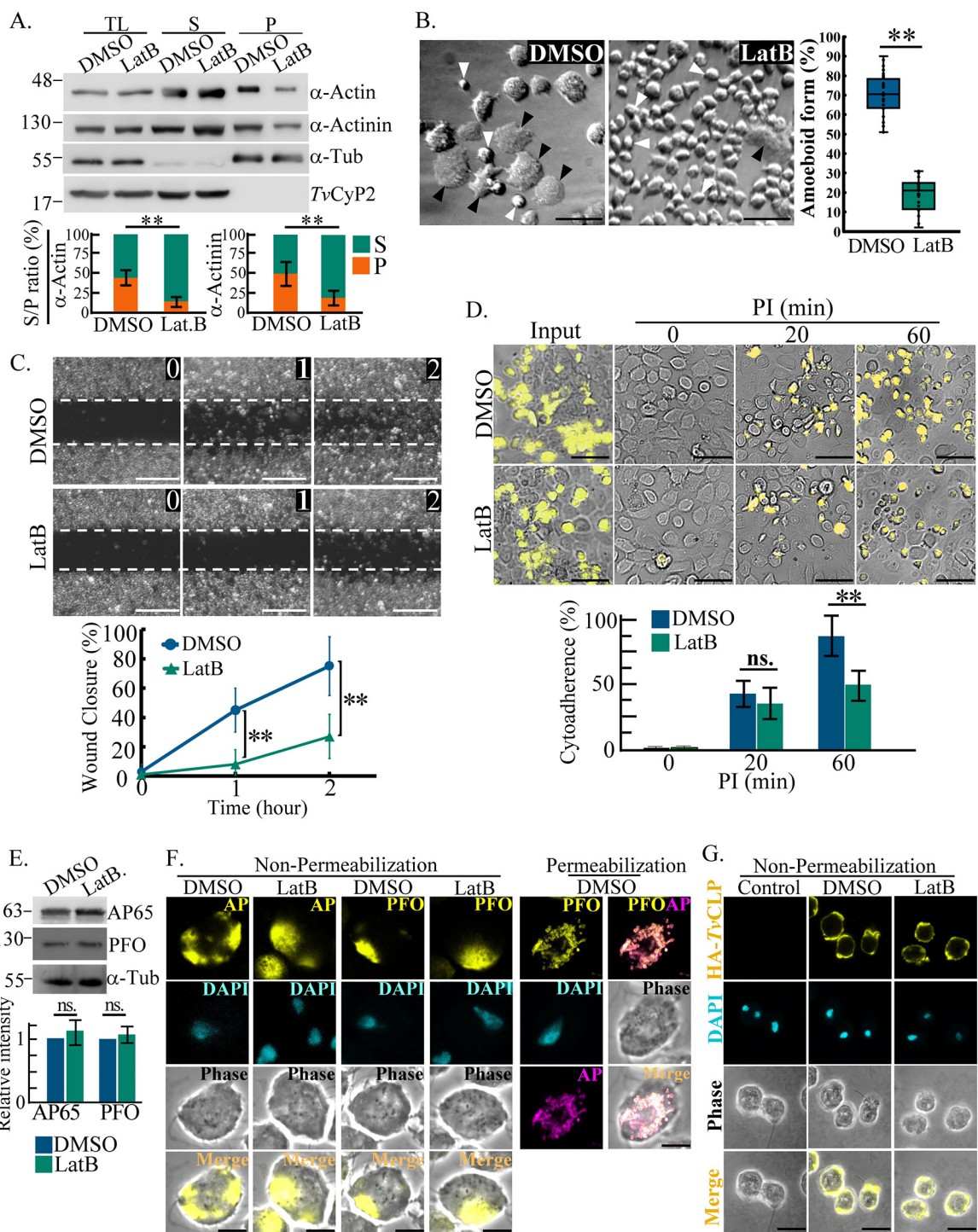

**FIG 3** Dysregulation of cytoskeleton-dependent behaviors in *T. vaginalis*. TH17 adherent trophozoites pretreated with DMSO or LatB were sampled for various assays. (A) Total lysates (TL) or protein lysates from actin fractionation were subjected to Western blotting. The signal ratios of the indicated proteins in the supernatant (S) versus the pellet (P) were measured, as shown in the bar graphs. (B) Trophozoite morphology was observed by phase-contrast microscopy. The proportion of trophozoites in the amoeboid form was measured in 600 trophozoites from 12 random microscopic fields, as shown in the box-and-whisker plot. The black and white arrowheads indicate the representative amoeboid and flagellate forms of trophozoites, respectively. Bars, 20 μm. (C) For the wound-healing assay, representative images were captured at 0, 1, and 2 h. The wound closure rate was measured as a percentage of the wound recovery area at the indicated time points. The white dashed lines depict the initial wound edge. Bars, 200 μm. (D) In the binding assay, conditional trophozoites were cocultured with hVECs for the times indicated. The ratio of trophozoites binding versus the input was measured at different time points postinfection (PI), as shown in the bar graph. Bars, 100 μm. (E) Total lysates from conditional trophozoites were subjected to Western blotting. (F) Fixed trophozoites with or without permeabilization were stained by anti-PFO and anti-AP65 antibodies for IFAs. Bars, 5 μm. (G) Nontransgenic control and transgenic trophozoites overexpressing HA-

chaperones (5%), membrane trafficking and transporter proteins (10%), protein binding or modification enzymes (7%), DNA/RNA regulation and translation proteins (17%), metabolism enzymes (37%), and uncharacterized proteins (2%) (Fig. S3B). The top five most abundant proteins identified in the immunoprecipitation (IP) proteome are listed in Fig. S3C. An F-actin CP subunit $\alpha$ homolog, referred to as *Tv*FACP$\alpha$ (TVAG_470230), had an emPAI score of ~9.7, supporting a strong protein-protein interaction between *Tv*FACP$\alpha$ and *Tv*Actin. The *in silico* protein sequence analysis revealed that *Tvfacp$\alpha$* encodes 267 amino acids with a molecular weight of 29.1 kDa and a pI value of 5.43 and shows 17% identity and 63% similarity to higher eukaryotic CP$\alpha$ (Fig. S3D). *Tv*FACP$\alpha$ contains a conserved actin-binding domain at the C terminus spanning amino acids 237 to 261. By using a phosphorylation site prediction algorithm (NetPhos 3.1 generic phosphorylation prediction [https://Services.healthtech.dtu.dk/service.php?NetPhos-3.1]), Ser2, Ser46, Ser88, Ser106, and Ser223 were predicted to be CKII phosphorylation sites. The sequence $^2$SESE$^5$ fits the putative CKII phosphorylation motif (pS/pTDXE) possibly recognized by a phospho-CKII substrate antibody. Basic Local Alignment Search Tool (BLAST) analysis identified two CP$\alpha$-homologous proteins (TVAG_470230 and TVAG_212270) in the TrichDB database with 32% sequence similarity (Fig. S4), but whether they are functionally redundant in this parasite remains to be studied.

***In vitro* functional analysis of *Tv*FACP$\alpha$.** To examine the role of *Tv*FACP$\alpha$ in *Tv*Actin polymerization, His-*Tv*FACP$\alpha$, His-$\Delta$237–261, glutathione *S*-transferase (GST), GST-*Tv*Actin, and tagless monomeric recombinant *Tv*Actin (G-r*Tv*Actin) were purified for *in vitro* assays (Fig. S5A). The polymerization of r*Tv*Actin (F-r*Tv*Actin) was evaluated by fluorescence changes within the initial 20 min of a pyrene fluorescence assay. A signal was triggered with 2 $\mu$M G-r*Tv*Actin and increased with increasing G-r*Tv*Actin concentrations (Fig. 4A). However, the reaction was dose-dependently reduced by His-*Tv*FACP$\alpha$ but was less affected by His-$\Delta$237–261, indicating that *Tv*FACP$\alpha$ inhibits actin assembly (Fig. 4B). Negative-staining transmission electron microscopy (TEM) revealed that the filaments polymerized by 4 $\mu$M G-r*Tv*Actin for 10 min were reduced in length from 6.56 $\pm$ 2.1 $\mu$m to 3.42 $\pm$ 1.21 $\mu$m in the presence of His-*Tv*FACP$\alpha$ (Fig. 4C [magnification, $\times$20,000] and Fig. S5B [magnification, $\times$40,000]), supporting the function of *Tv*FACP$\alpha$ in inhibiting actin polymerization. Total internal reflection fluorescence (TIRF) microscopy (Fig. 4D) showed that 6 $\mu$M G-r*Tv*Actin was polymerized into filaments at an average assembly rate of 11.3 $\pm$ 2.8 subunits/s, but this was reduced to 5.9 $\pm$ 2.1 subunits/s by His-*Tv*FACP$\alpha$. Meanwhile, *Tv*FACP$\alpha$ was transiently detected at the barbed end of the growing filament (Fig. 4Da and b) to repress elongation (Fig. 4D, kymographs I and II). The assembly rates with different concentrations of r*Tv*Actin were measured (Video S3 and Fig. S6) to obtain an assembly rate constant of 2.87 $\pm$ 0.6 subunits/$\mu$M $\cdot$ s, a disassembly rate constant of 6.07 $\pm$ 2.2 subunits/s, and a critical concentration of 2.1 $\pm$ 0.5 $\mu$M, which changed to 1.45 $\pm$ 0.74 subunits/$\mu$M $\cdot$ s, 3.2 $\pm$ 1.7 subunits/s, and 2.2 $\pm$ 0.5 $\mu$M, respectively, in the presence of His-*Tv*FACP$\alpha$ (Fig. 4E). In contrast to F-buffer and His-$\Delta$237–261, His-*Tv*FACP$\alpha$ was cosedimented with F-r*Tv*Actin after ultracentrifugation (Fig. 4F). Since His-*Tv*FACP$\alpha$ reduced the overall assembly rate regardless of the binding of F-r*Tv*Actin (Fig. 4D, kymograph II), we examined whether *Tv*FACP$\alpha$ also interferes with assembly via binding to G-actin. His-*Tv*FACP$\alpha$ was copulled down with monomeric GST-*Tv*Actin beads (Fig. 4G) and bound to a G-r*Tv*Actin-coated microplate (Fig. S7), demonstrating the atypical G-actin-binding activity of *Tv*FACP$\alpha$.

***Tv*FACP$\alpha$ represses F-actin assembly in *T. vaginalis*.** The anti-*Tv*FACP$\alpha$ antibody identified an ~30-kDa protein band in the total lysate from TH17 trophozoites by Western blotting (Fig. 5A) and colocalized with TRITC-phalloidin, with a Pearson corre-

**FIG 3** Legend (Continued)

*Tv*CLP were stained by anti-HA antibody for IFAs under nonpermeabilization conditions. Bars, 10 $\mu$m. All micrographs were captured in a single z-slice. All experiments were performed with three biological repeats ($n = 3$). Data in the bar graphs and line chart are presented as means $\pm$ SD. Statistical significance for each group of data was analyzed by Student's *t* test, as indicated ($n = 3$) (\*\*, $P < 0.01$; ns, no significance).

**TABLE 1** List of *Tv*Actin-interacting proteins identified by LC-MS/MS[a]

| Accession no. | Score | Mass (Da) | emPAI score | Description |
|---|---|---|---|---|
| Chaperones | | | | |
| A2DS85 | 56 | 58,182 | 0.15 | T-complex protein 1 subunit delta; TVAG_066690 |
| A2E9D9 | 62 | 58,553 | 0.15 | Chaperonin subunit $\alpha$1 CCT$\alpha$, putative; TVAG_364270 |
| DNA/RNA-binding or -regulatory proteins | | | | |
| A2DHC5 | 37 | 15,204 | 0.31 | Histone H2A; TVAG_021440 |
| A2ELI6 | 46 | 11,528 | 0.42 | HTH Myb-type domain-containing protein; TVAG_257520 |
| A2D755 | 78 | 127,201 | 0.1 | DEAD/DEAH box helicase family protein; TVAG_119080 |
| Cytoskeletal proteins | | | | |
| A2FE30 | 541 | 29,551 | 9.69 | F-actin-capping protein subunit $\alpha$; TVAG_470230 |
| A2E0V9 | 104 | 46,956 | 0.3 | Actin-like protein 3, putative; TVAG_371880 |
| A2E755 | 69 | 533,676 | 0.02 | Dynein heavy chain family protein; TVAG_006480 |
| A2EIJ3 | 43 | 48,424 | 0.19 | Coronin; TVAG_124870 |
| A2DC16 | 208 | 50,493 | 0.51 | Tubulin $\beta$-chain; TVAG_008680 |
| A2EGW8 | 73 | 515,283 | 0.02 | Dynein heavy chain family protein; TVAG_497260 |
| A2GKR2 | 363 | 26,669 | 5.37 | Actin (fragment); TVAG_534990 |
| P90623 | 659 | 42,154 | 9.49 | Actin; TVAG_337240 |
| A2DKH3 | 163 | 106,648 | 0.21 | $\alpha$-Actinin, putative; TVAG_190450 |
| Membrane traffic proteins | | | | |
| A2EV08 | 90 | 85,200 | 0.1 | Clathrin and VPS domain-containing protein; TVAG_369030 |
| Metabolite interconversion enzymes | | | | |
| A2DSX4 | 80 | 107,447 | 0.21 | $\alpha$-1,4-Glucan phosphorylase; TVAG_348330 |
| A2EBX0 | 218 | 44,060 | 0.76 | Succinate-CoA ligase (ADP-forming) subunit $\beta$, mitochondrial; TVAG_259190 |
| A2FR66 | 116 | 35,697 | 0.26 | 6-Phosphofructokinase; TVAG_496160 |
| A2E9H3 | 37 | 47,022 | 0.19 | Pyrophosphate-fructose 6-phosphate 1-phosphotransferase 2; TVAG_364620 |
| A2FVK7 | 224 | 44,039 | 1.12 | Succinate-CoA ligase (ADP-forming) subunit $\beta$, mitochondrial; TVAG_144730 |
| A2DM03 | 169 | 34,697 | 0.61 | 6-Phosphofructokinase; TVAG_462920 |
| A2FKA7 | 82 | 34,780 | 0.27 | 6-Phosphofructokinase; TVAG_293770 |
| A2EM29 | 503 | 39,758 | 6.2 | Glyceraldehyde-3-phosphate dehydrogenase; TVAG_475220 |
| Q27088 | 86 | 129,430 | 0.1 | Pyruvate:ferredoxin oxidoreductase A; TVAG_198110 |
| A2EAJ8 | 98 | 42,785 | 0.47 | Malic enzyme; TVAG_491670 |
| A2DM76 | 95 | 38,889 | 0.11 | Thymidine kinase; TVAG_083490 |
| A2D987 | 238 | 44,049 | 1.33 | Succinate-CoA ligase (ADP-forming) subunit $\beta$, mitochondrial; TVAG_183500 |
| A2DFT9 | 198 | 35,079 | 1.02 | 6-Phosphofructokinase; TVAG_391760 |
| A2F259 | 111 | 109,764 | 0.16 | Amylomaltase; TVAG_154680 |
| A2D7H3 | 94 | 110,021 | 0.08 | Amylomaltase; TVAG_120280 |
| Protein-modifying enzymes | | | | |
| A2EPF2 | 49 | 95,046 | 0.09 | Proteasome/cyclosome repeat family protein; TVAG_286380 |
| Protein-binding activity modulators | | | | |
| A2EB65 | 53 | 40,705 | 0.22 | G-protein $\alpha$ subunit, putative; TVAG_274750 |
| A2EJL0 | 36 | 24,328 | 0.18 | IBD domain-containing protein; TVAG_197940 |
| Translation proteins | | | | |
| A2E4D0 | 39 | 40,574 | 0.11 | Ribosomal protein, putative; TVAG_128790 |
| A2DSF6 | 55 | 48,559 | 0.19 | Elongation factor 1$\alpha$; TVAG_067400 |
| A2ECS2 | 117 | 94,235 | 0.36 | Tr-type G-domain-containing protein; TVAG_276410 |
| A2DSV0 | 99 | 28,352 | 0.34 | Ribosomal protein S3Ae, putative; TVAG_348090 |
| Transporters | | | | |
| A2FS41 | 70 | 49,238 | 0.18 | V-ATPase_H_C domain-containing protein; TVAG_262750 |
| A2ED49 | 68 | 68,309 | 0.2 | H$^+$-transporting two-sector ATPase; TVAG_420260 |
| A2ES57 | 50 | 55,789 | 0.41 | Vacuolar proton pump subunit B; TVAG_453110 |
| Uncharacterized proteins | | | | |
| A2E2D0 | 59 | 149,814 | 0.06 | Uncharacterized protein; TVAG_098000 |

[a]The proteins identified by mass spectrometry with emPAI values of >0.25 or peptides specific in the immunoprecipitant of HA-*Tv*Actin are listed. HTH, helix-turn-helix; VPS, vacuolar protein sorting; IBD, initiator binding domain. The proteins with specific accession numbers were searched from TrichDB (https://trichdb.org/trichdb/app).

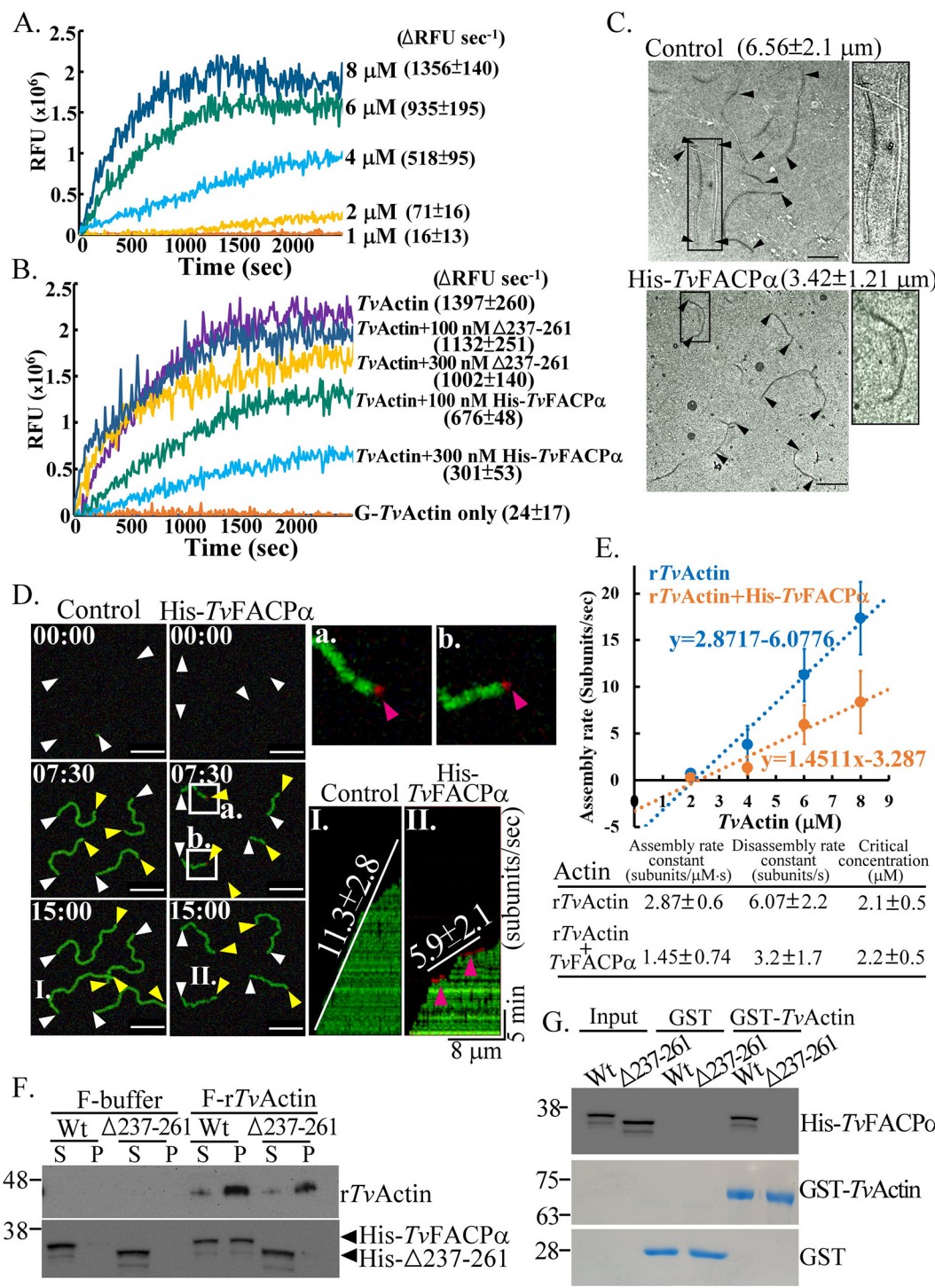

**FIG 4** *In vitro* functional analysis of *Tv*FACPα. Recombinant His-*Tv*FACPα, His-Δ237–261, GST, GST-*Tv*Actin, and tagless r*Tv*Actin were produced for *in vitro* polymerization assays (see Fig. S5A in the supplemental material). (A) The relative fluorescence unit (RFU) changes with 1, 2, 4, 6, and 8 $\mu$M r*Tv*Actin polymerization were monitored by using a fluorometer. (B) The RFU changes with 6 $\mu$M G-r*Tv*Actin polymerization with 100 or 300 nM His-*Tv*FACPα or His-Δ237–261 were monitored over time. In panels A and B, the data were zeroed by subtracting the basal fluorescence from all fluorescence values. The polymerization velocity was estimated by the RFU change within the initial 1,200-s reaction, as shown in parentheses. G-r*Tv*Actin only was detected as a negative control. (C) Four micromolar G-r*Tv*Actin polymerized for 10 min in the presence or absence of 200 nM His-*Tv*FACPα was visualized by negative-staining TEM. Arrowheads label the ends of actin filaments. Bars, 1 $\mu$m. The boxed regions are magnified on the right side of each panel. (D) Time-lapse images with 6 $\mu$M G-r*Tv*Actin polymerization with 300 nM BSA control or His-*Tv*FACPα were recorded by TIRF microscopy. The white and yellow arrowheads indicate the pointed and barbed ends of elongating filaments, respectively. Bars, 10 $\mu$m. The regions in panels a and b were magnified to show Alexa Fluor 555-His-*Tv*FACPα (magenta arrowheads) binding at the

lation coefficient value of 0.86 (Fig. 5B and Fig. S8A). When HA-*Tv*FACPα and HA-Δ237–261 were overexpressed in the cytoplasm (Fig. 5C), an uneven cytoplasmic distribution of α-actin with sporadic patches underneath the cell membrane was observed in the nontransgenic parasites and the HA-Δ237–261 mutant. Increasing numbers of tiny α-actin puncta were detected in the HA-*Tv*FACPα-overexpressing transfectant, indicating that *Tv*FACPα overexpression may alter actin organization in this parasite.

HA-*Tv*FACPα or HA-Δ237–261 was overexpressed at a level ~5-fold higher than that of the endogenous form, and the former inhibited endogenous *Tv*FACPα expression in the transfectant (Fig. 5D), suggesting that a feedback pathway maintains *Tv*FACPα levels. The expression of α-actin or α-actinin was unchanged between transfectants (Fig. 5D and Fig. S8B). Fractionation revealed that ~45% F-actin cosedimented with ~25% *Tv*FACPα in the nontransgenic TH17 control. In HA-*Tv*FACPα transfectants, the F-actin level was reduced to ~25%, but the level of cosedimented HA-*Tv*FACPα was ~2-fold higher than that of the endogenous form in the nontransfectant. In the HA-Δ237–261 mutant, the F-actin ratio was slightly higher, but the level of cosedimented HA-Δ237–261 was lower than that of the nontransfectant; therefore, *Tv*FACPα may repress actin polymerization. A similar trend was observed for α-actinin (Fig. S8B). By immunoprecipitation, the levels of α-actin and α-actinin coprecipitated with HA-*Tv*FACPα were much lower in the HA-Δ237–261 mutant (Fig. 5E and Fig. S8C), indicating that actin-binding activity is essential for *Tv*FACPα to inhibit actin assembly.

**The function of *Tv*FACPα in actin polymerization is regulated by CKII signaling.** More *Tv*FACPα and α-actin were expressed in adherent TH17 isolates, but less *Tv*FACPα cosedimented with F-actin (Fig. S9). The immunostaining of α-actin was distinct between the flagellate and amoeboid trophozoites (Fig. 2B), with equal amounts of *Tv*FACPα, α-actin, and α-actinin being detected in the total lysates (Fig. 6A and Fig. S10A). In amoeboid trophozoites, the F-actin ratio was 2-fold higher, but the level of *Tv*FACPα cosedimented with F-actin was 2-fold lower than that of the flagellate form (Fig. 6A), indicating that the amoeboid form exhibits more active F-actin polymerization and less *Tv*FACPα binding α-actin.

*Tv*FACPα Ser2 was previously predicted to be a CKII phosphorylation site (Fig. S3D) potentially recognized by a phospho-motif (pS/pTDXE)-specific antibody, referred to as *Tv*FACP(pS2). When *Tv*FACPα was equally immunoprecipitated from the trophozoites, more *Tv*FACP(pS2) but less α-actin and α-actinin were co-pulled down from the amoeboid trophozoites than from the flagellate form (Fig. 6B and Fig. S10B). Ser2 may be hyperphosphorylated in the amoeboid trophozoites, reducing α-actin or α-actinin binding. To confirm the role of Ser2 phosphorylation in the complex formation of *Tv*FACPα and α-actin, hypophosphorylation S2A or hyperphosphorylation S2D mutants were introduced into TH17 trophozoites for fractionation and immunoprecipitation. The overall levels of α-actin and α-actinin were unchanged between transfectants. The overexpression of both HA-*Tv*FACPα and S2A repressed F-actin levels, with higher levels of HA-*Tv*FACPα or S2A being cosedimented in the F-actin fraction (Fig. 6C and Fig. S10C). In contrast, similar levels of F-actin were detected in the nontransfectant and S2D mutants, but the level of S2D cosedimented in the F-actin fraction was lower than that of HA-*Tv*FACPα (Fig. 6D and Fig. S10D). Similar results were obtained for α-actinin. Furthermore, the α-actin signals coimmunoprecipitated from the S2A and S2D mutants were 3-fold higher and 70% lower, respectively, than those of HA-*Tv*FACPα (Fig. 6E and

**FIG 4** Legend (Continued)

barbed end of F-r*Tv*Actin. The respective kymographs (I and II) show the assembly rates of filaments I and II in the left panel. (E) A plot of the r*Tv*Actin assembly rate versus the concentration was generated from 25 independent growing filaments, from each of two protein preparations, to obtain the assembly rate constant, the disassembly rate constant, and the critical concentration. (F) Polymerized F-r*Tv*Actin was incubated with His-*Tv*FACPα or His-Δ237–261 for cosedimentation assays and Western blotting. (G) Equimolar concentrations of His-*Tv*FACPα and His-Δ237–261 were reacted with GST and monomeric GST-*Tv*Actin immobilized on Sepharose beads for pulldown assays and Western blotting. The assays were performed with three biological repeats ($n = 3$) unless specified otherwise. Data are presented as means ± SD. Significant differences in the $P$ values for each group of data were statistically analyzed by Student's $t$ test ($n = 3$). Wt, wild type.

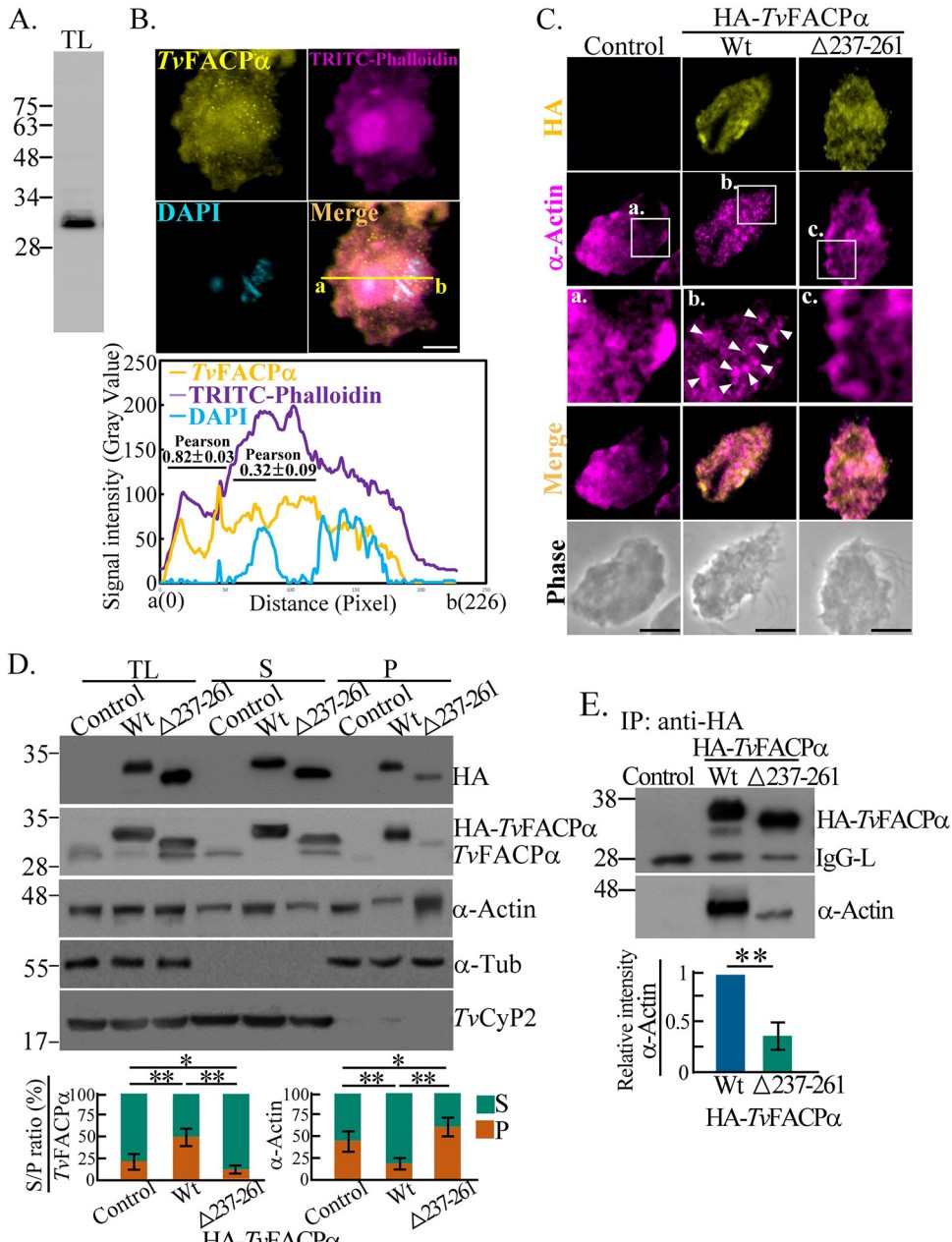

**FIG 5** *Tv*FACPα binds actin to block F-actin assembly in *T. vaginalis*. (A) The total lysate from TH17 trophozoites was subjected to Western blotting with an anti-*Tv*FACPα antibody. (B) TH17 trophozoites cultivated on a glass slide were costained with anti-*Tv*FACPα antibody and TRITC-phalloidin. Bar, 2 μm. The signal was assessed by plot profile analysis to display the intensity distributions between sites a and b on the yellow line. The overall colocalization of phalloidin with *Tv*FACPα or DAPI was evaluated by Pearson correlation coefficients, as shown in Fig. S8A in the supplemental material. Subcellular colocalization was assessed as indicated in the bottom plot (pseudopodia, 0.82 ± 0.03; juxtanuclear, 0.32 ± 0.09). Data are presented as means ± SD. (C) IFA of nontransgenic control and transgenic TH17 trophozoites overexpressing HA-*Tv*FACPα or HA-Δ237–261 was performed by using anti-HA and anti-α-actin. Magnified images of the boxed regions are shown in panels a to c. The α-actin puncta are indicated by white arrowheads. Bars, 5 μm. The images in panels B and C were captured in a single z-slice. (D) Total lysates and actin fractionations from nontransgenic control and transgenic TH17 trophozoites were examined by Western blotting. The ratio of the indicated protein signal in the supernatant fraction (S) to that in the pellet fraction (P) (S/P ratio) was analyzed, as shown in the bar graph. (E) The total lysates from the trophozoites from panel D were immunoprecipitated with an anti-HA antibody for Western blotting. The relative intensity of the indicated protein signal was normalized to the intensity of the input lysates, as shown in the bar graph. All assays were performed with three biological repeats ($n = 3$). Data in the bar graphs are presented as means ± SD. Statistical significance for each group of data was analyzed by Student's *t* test, as indicated ($n = 3$) (**, $P < 0.01$; *, $P < 0.05$).

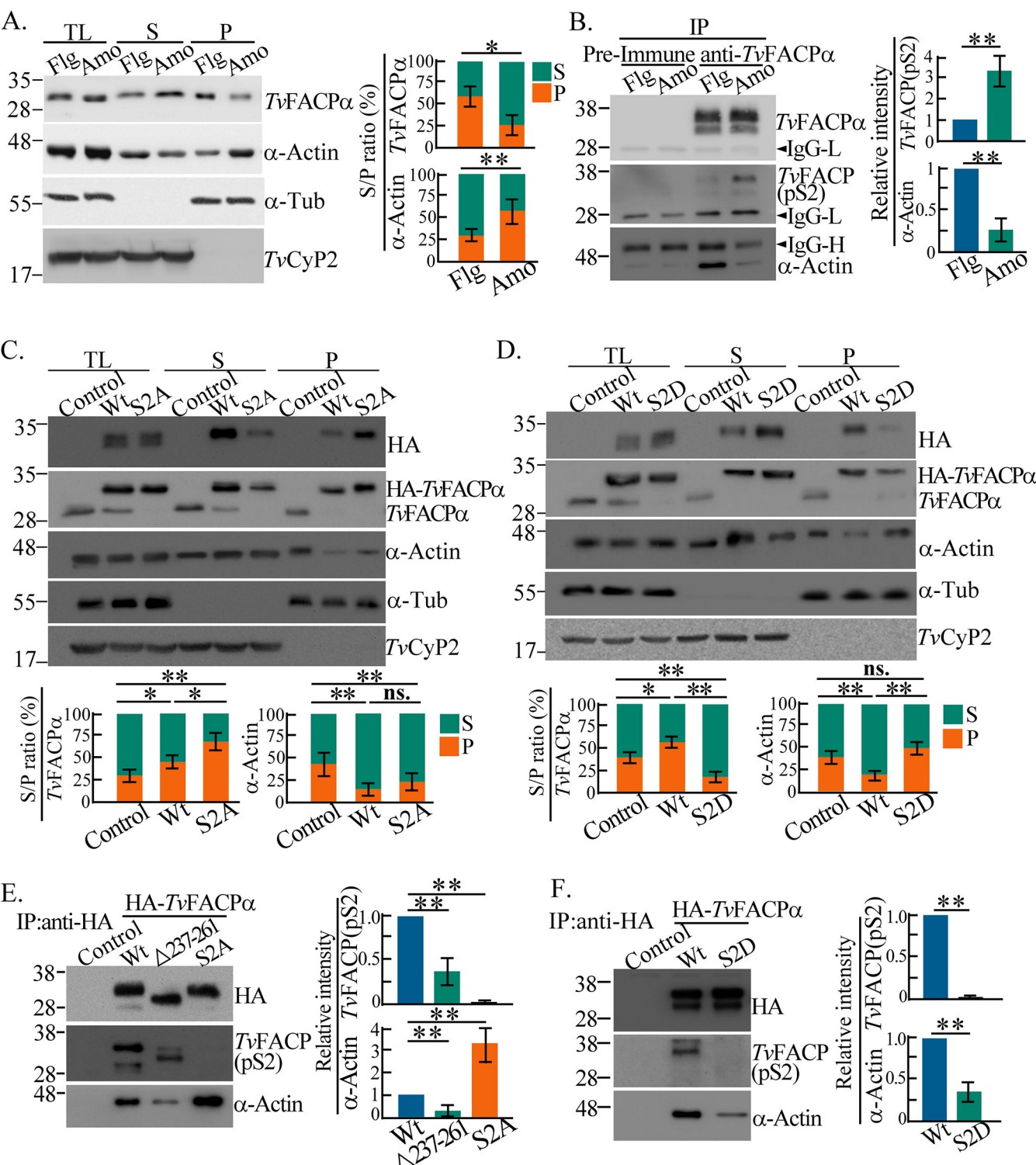

**FIG 6** Ser2 phosphorylation regulates *Tv*FACPα actin binding. (A) Total lysates from TH17 trophozoites in the flagellate (Flg) and amoeboid (Amo) forms were fractionated for Western blotting. The ratio of the indicated protein signal in the pellet (P) to that in the supernatant (S) was quantified, as shown in the bar graph. (B) The immunoprecipitants from the total lysates from panel A were examined by Western blotting using anti-*Tv*FACPα antibody. The relative signal intensities of the indicated proteins were quantified, as shown in the bar graph. (C and D) Total lysates from nontransgenic control or TH17 trophozoites overexpressing HA-*Tv*FACPα and S2A (C) or S2D (D) were fractionated for Western blotting. The ratios of the indicated protein signals from the pellet fraction (P) to those in the supernatant fraction (S) were analyzed, as shown in the bar graph. (E and F) The total lysates from trophozoites overexpressing HA-*Tv*FACPα and S2A (E) or S2D (F) were immunoprecipitated by an anti-HA antibody for Western blotting. The relative intensities of the indicated protein signals were quantified, as shown in the bar graphs. All assays were performed with three biological repeats (*n* = 3). Data are presented as means ± SD. Statistical significance for each group of data was measured by Student's *t* test, as indicated (*n* = 3) (\*\*, *P* < 0.01; \*, *P* < 0.05; ns, no significance).

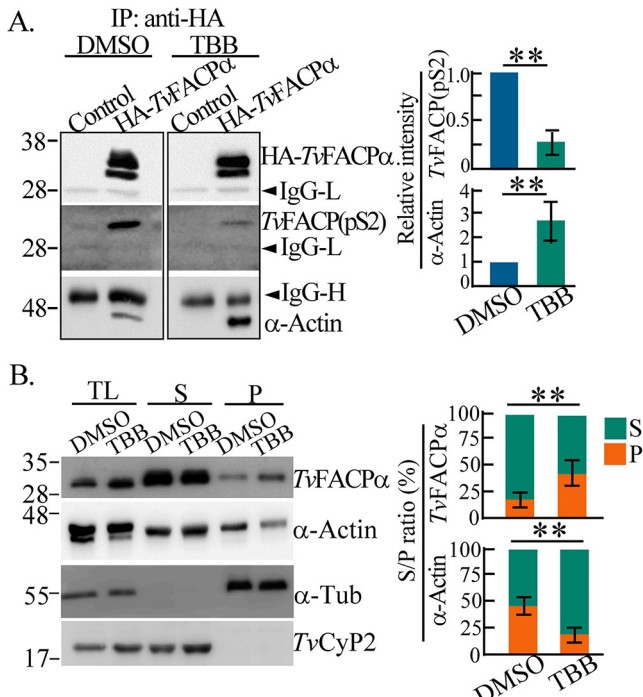

**FIG 7** CKII signaling regulates *Tv*FACPα actin binding. (A) The total lysates from nontransgenic control or HA-*Tv*FACPα-overexpressing TH17 trophozoites with DMSO or TBB treatment were sampled for Western blotting (see Fig. S13A in the supplemental material) or immunoprecipitation by anti-HA antibody. The relative intensities of the signals were quantified, as shown in the bar graphs. (B) TH17 trophozoites treated with DMSO or TBB were fractionated for Western blotting. The α-actinin signal is shown in Fig. S13B. The ratio of the indicated protein signal in the pellet fraction (P) to that in the supernatant fraction (S) was quantified, as shown in the bar graph. All assays were performed with three biological repeats ($n = 3$). Data are presented as means ± SD. Statistical significance for each group of data was measured by Student's $t$ test, as indicated ($n = 3$) (**, $P < 0.01$).

F and Fig. S10E and F), with a low α-actin signal coimmunoprecipitated with the HA-Δ237–261 mutant, implying that Ser2 phosphorylation regulates the actin-binding activity of *Tv*FACPα. Meanwhile, the low intensity of the *Tv*FACP(pS2) signal detected from the HA-Δ237–261 mutant implies that the integrity of the actin-binding domain might be important for Ser2 phosphorylation. Ser2 phosphorylation is a major signal for the dissociation of *Tv*FACPα and α-actin. However, *Tv*FACP(pS2) was undetectable in the immunoprecipitant of the S2A or S2D mutant (Fig. 6E and F) or reduced by phosphatase treatment (Fig. S11A), supporting the signal specificity of *Tv*FACP(pS2).

To verify the role of CKII signaling in Ser2 phosphorylation, TH17 trophozoites overexpressing HA-*Tv*FACPα were treated with 4,5,6,7-tetrabromobenzotriazole (TBB), a CKII inhibitor (39). The *Tv*FACP(pS2) signal in the immunoprecipitant was reduced by TBB, with a 50% inhibitory concentration (IC$_{50}$) of 0.58 μM (Fig. S11B). However, TBB did not affect parasite viability (Fig. S11C). Several *T. vaginalis* CKIIα (*Tv*CKIIα) proteins sharing consensus ATP- and TBB-binding sites with mouse CKIIα were identified in TrichDB (Fig. S12), supporting TBB's efficacy against this parasite. When HA-*Tv*FACPα was equally immunoprecipitated from the trophozoites, TBB decreased the *Tv*FACP(pS2) signals but increased the α-actin signals (Fig. 7A and Fig. S13A). When the expression levels of *Tv*FACPα, α-actin, and α-actinin remained constant in TH17 trophozoites, F-actin in TBB-treated parasite was inhibited to one-third of the basal level, and the level of *Tv*FACPα cosedimented with F-actin was 3-fold higher than that of the DMSO control (Fig. 7B and Fig. S13B). Together, CKII-dependent Ser2 phosphorylation triggers the dissociation of *Tv*FACPα and α-actin to evoke actin polymerization.

**Function of *Tv*FACPα in the morphogenesis and cytoadherence of *T. vaginalis*.** To examine the effect of Ser2 phosphorylation on cytoskeleton behaviors, the morphogenesis of transfectants was monitored. Trophozoites overexpressing HA-*Tv*FACPα and

S2A reduced amoeboid morphogenesis to ~20%, compared to ~70% in the nontransgenic control, whereas it was restored to ~70% in the HA-Δ237–261 and S2D mutants (Fig. 8A). TBB treatment also reduced TH17 morphogenesis from ~80% in the DMSO control to ~30%. Notably, TBB-inhibited morphogenesis was abolished in the S2D transfectant, suggesting that CKII-dependent Ser2 phosphorylation in *Tv*FACPα is crucial for *T. vaginalis* amoeboid morphogenesis (Fig. 8B). The cytoadherence of various transfectants was monitored over time. Sixty minutes after infection, ~100% cytoadherence in the nontransgenic TH17 strain was reduced to ~40% in the HA-*Tv*FACPα and S2A transfectants and was increased to ~80% in the HA-Δ237–261 and S2D transfectants (Fig. 8C). TBB treatment also significantly reduced TH17 cytoadherence 60 min after infection, but this was abrogated in the S2D transfectant (Fig. 8D). Notably, the transfectants or TBB treatment did not affect cytoadherence during the initial 20 min of infection (Fig. 8C and D), consistent with our above-described observation that LatB perturbed cytoadherence only 60 min after infection (Fig. 3D). These data strongly support that CKII-dependent Ser2 phosphorylation regulates the function of *Tv*FACPα in cytoskeleton-mediated *T. vaginalis* morphogenesis and consequent cytoadherence.

**Function of *Tv*FACPα in amoeboid migration.** Since cytoskeletal disorder retarded the morphogenesis and reduced the adherence activity of *T. vaginalis* (Fig. 3 and 8), conditional trophozoites were cultured to a monolayer in a T25 flask for a wound-healing assay. The wound recovery rate was decreased in the HA-*Tv*FACPα transfectant, which was partially reversed in the HA-Δ237–261 mutant, showing that *Tv*FACPα actin-binding activity essential for reducing amoeboid migration (Fig. 9A). Also, the wound recovery rate in TH17 trophozoites was decreased by TBB to a level similar to that of the HA-*Tv*FACPα transfectant. In contrast, the wound closure rate in the S2D mutant was similar to that of the nontransgenic control and was not influenced by TBB treatment (Fig. 9B), revealing that the S2D mutant counteracts TBB's inhibitory effect on amoeboid migration. This observation indicates the critical role of CKII-dependent Ser2 phosphorylation in *Tv*FACPα-regulated amoeboid migration.

**_Tv_FACPα regulates motility switching in _T. vaginalis_.** We tested whether parasite motility is changed with the morphology transition using a transwell system (Fig. 9C). The relative glyceraldehyde-3-phosphate dehydrogenase (GAPDH) or α-tubulin signals detected by Western blotting indicate the relative amount of migratory trophozoites between the bottom wells and the top inserts. GAPDH or α-tubulin expression levels between the input trophozoites and the HA signals between the input transfectants were equal. Focusing the GAPDH signals from the bottom well of the 30-min transwell plate, the level of HA-*Tv*FACPα was higher but those of the HA-Δ237–261 and S2D mutants were lower than that of the nontransgenic control, revealing that more trophozoites with HA-*Tv*FACPα overexpression migrated to the bottom well in a short time (Fig. 9D). As observed by microscopy, the trophozoites in the bottom well displayed the morphology of the free-swimming flagellate form (Fig. S14A), suggesting that HA-*Tv*FACPα overexpression may retain the parasite in the flagellate form with faster movement driven by motile flagella. When TBB inhibited Ser2 phosphorylation in *Tv*FACPα (Fig. 7A), the GAPDH or α-tubulin signals from the TBB-treated trophozoites in the bottom well were higher than those from the DMSO-treated trophozoites (Fig. 9E), showing that more flagellates migrated into the bottom well. A similar tendency was also observed by directly counting migrating trophozoites in the bottom wells (Fig. S14B and C). *Tv*FACPα actin-binding activity regulated by Ser2 phosphorylation involves parasite motility conversion. Together, *Tv*FACPα Ser2 hypophosphorylation retarded amoeboid migration in the adhered trophozoites but expanded the population of free trophozoites that rapidly moved via flagellar locomotion.

## DISCUSSION

*Tv*FACPα was identified as an actin-binding protein that suppresses actin polymerization. Furthermore, CKII-dependent signaling switches morphology and motility. These cytoskeleton behaviors are crucial for the parasite to colonize the host. In the human urogenital tract, the intermittent flushing action of body fluid generates a mechanical barrier to either impair or eliminate uropathogenic microbes; therefore, switching to the opportune motility

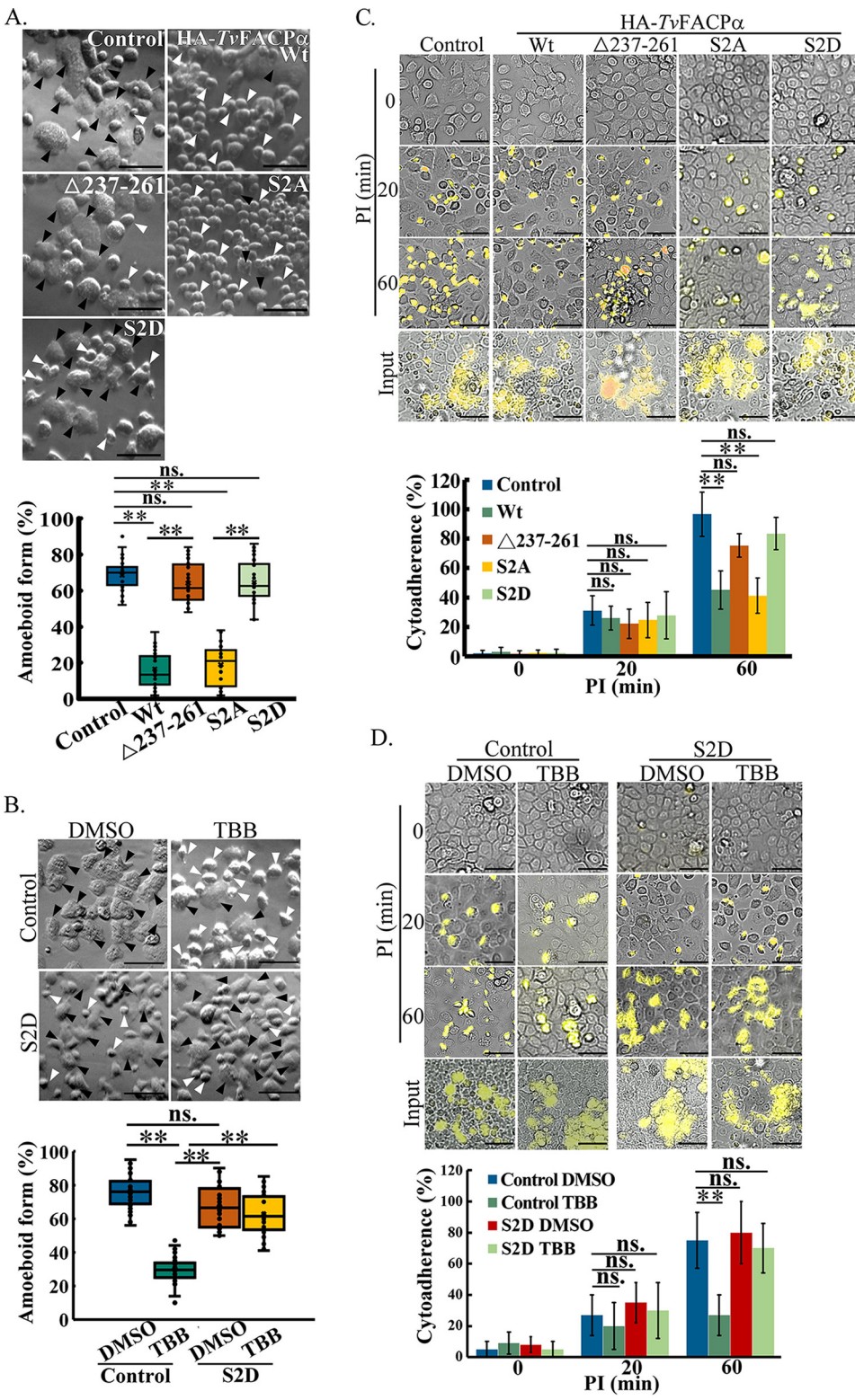

**FIG 8** *Tv*FACPα regulates actin-related morphogenesis and cytoadherence in *T. vaginalis*. (A and B) Nontransgenic control and transgenic TH17 trophozoites overexpressing HA-*Tv*FACPα, HA-Δ237–261, S2A, and S2D (A) and nontransgenic TH17 trophozoites and those overexpressing S2D with DMSO and TBB (B) were cultured on a glass slide for 1 h. The cell morphology was recorded by phase-contrast microscopy. The proportion of parasites in the amoeboid form was measured in ∼600 trophozoites from 12 microscopic fields, as shown in the box-and-whisker plots. The black and white arrowheads indicate representative amoeboid and flagellate forms of trophozoites, respectively. Bars, 20 μm. (C and D) For the cytoadherence binding assay, CFSE-labeled trophozoites overexpressing

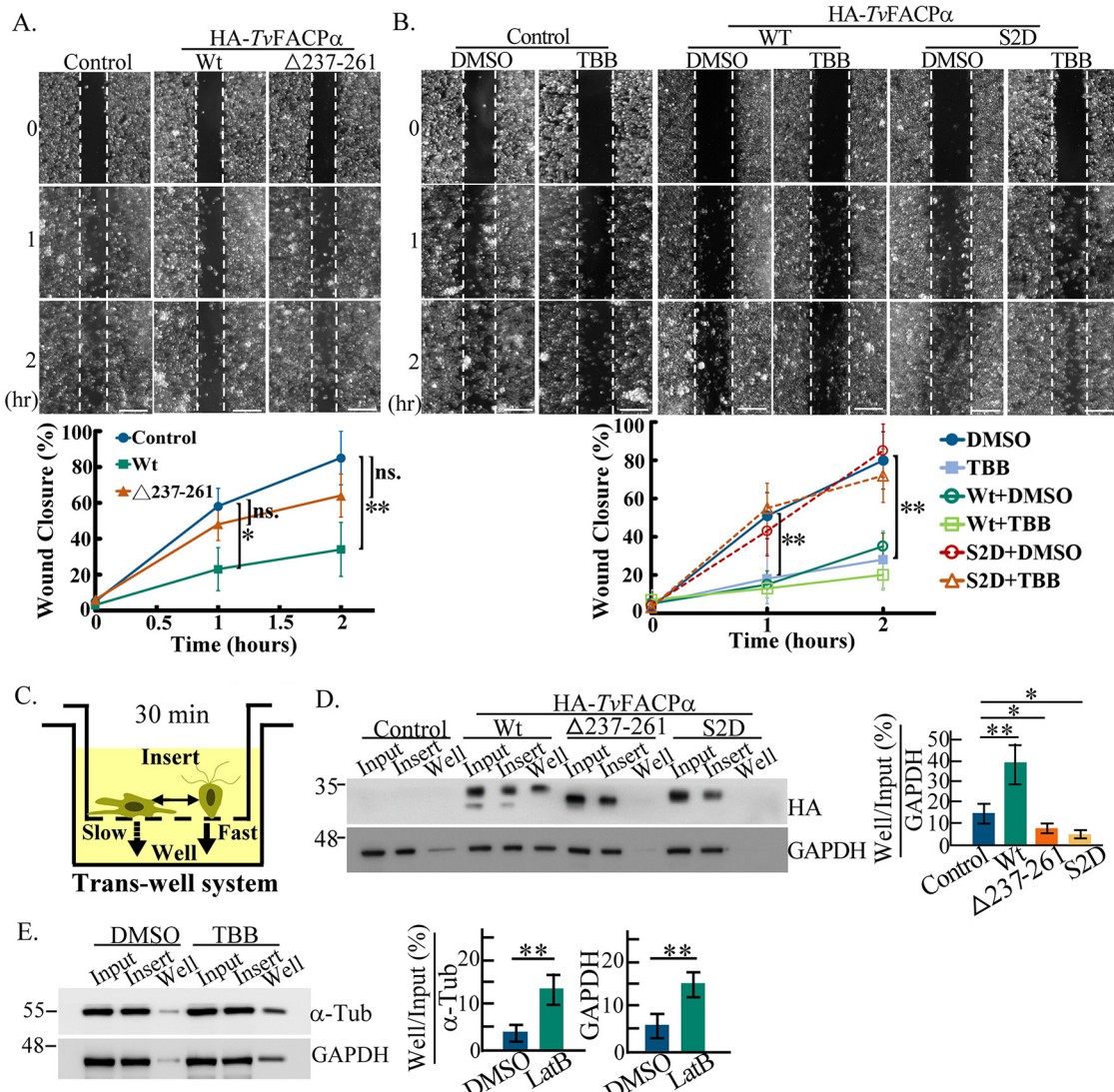

**FIG 9** *Tv*FACPα regulates amoeboid migration and motility switching of *T. vaginalis*. (A and B) The migrations of nontransgenic control and TH17 trophozoites overexpressing HA-*Tv*FACPα and HA-Δ237–261 (A) and those overexpressing HA-*Tv*FACPα and S2D with DMSO or TBB treatment (B) were evaluated by a scratch-wound-healing assay. Representative images showing wound closure were captured at 0, 1, and 2 h, and the closure rate was measured as a percentage of the wound recovery area at the indicated time points, as shown in the line charts. The white dashed lines mark the scratch wound boundaries. Bars, 200 μm. (C) Schematic diagram illustrating the working principle of a transwell system applied to assess migration. Within a short interval, free trophozoites swim by flagellar locomotion to pass through the boundary membrane faster than crawling by pseudopod migration. (D and E) The migrations of TH17 trophozoites overexpressing HA-*Tv*FACPα, HA-Δ237–261, and S2D (D) and TH17 trophozoites treated with DMSO and TBB (E) were evaluated by transwell assays. The relative intensities of the signals in the bottom well were evaluated by Western blotting and quantified, as shown in the bar graphs. All assays were performed with three biological repeats (*n* = 3). Data are presented as means ± SD. Statistical significance was measured by Student's *t* test, as indicated (*n* = 3) (**, *P* < 0.01; *, *P* < 0.05; ns, no significance).

mode to instantly counteract environmental challenges or physical defenses would be beneficial for *T. vaginalis* colonization (40).

The DNA sequences of the *Tvfacpα* gene from nonadherent and adherent *T. vaginalis* isolates show 100% identity (41); thus, the differential cytoskeleton behaviors between isolates are unlikely to be attributed to sequence polymorphisms in *Tv*FACPα.

**FIG 8** Legend (Continued)
HA-*Tv*FACPα and derived mutants (C) nontransgenic or S2D transgenic TH17 trophozoites pretreated with DMSO and TBB (D) were cocultured with hVECs for the indicated times. After the removal of unbound trophozoites, the ratios of those binding versus the input at different time points postinfection (PI) were measured, as shown in the bar graphs. Bars, 100 μm. All micrographs were captured in a single z-slice. All assays were performed with three biological repeats (*n* = 3). Data in the bar graphs are presented as means ± SD. Statistical significance for each group of data was analyzed by Student's *t* test, as indicated (*n* = 3) (**, *P* < 0.01; ns, no significance).

Bacterial expression systems have been used to produce human $\beta$-actin at 16°C by using a cold shock vector (42) or directly solubilizing *Escherichia coli*-synthesized slime mold actin with 0.2% Sarkosyl detergent (43), yielding soluble actin capable of myosin binding and polymerization, suggesting that bacterially expressed actin may be functional. The extremely slow dialysis of r*Tv*Actin into G-buffer may be critical for harvesting functional r*Tv*Actin. After rounds of polymerization/depolymerization coupled with ultracentrifugation, hundreds of micrograms of polymerization-competent r*Tv*Actin could be recovered from a 5-L *E. coli* culture. It is unclear whether the purification procedure alters the intrinsic properties of r*Tv*Actin, but its ability to form filaments was evidenced by TEM and TIRF microscopy, thereby elucidating the inhibitory effect of *Tv*FACP$\alpha$.

r*Tv*Actin polymerizes into F-actin *in vitro* at a critical concentration of 2 $\mu$M, which is slightly lower than that of 4 $\mu$M measured for Act1 of the malaria parasite *Plasmodium falciparum* (*Pf*Act1) but an order of magnitude higher than that of mammalian skeletal actin (44). Because the D-loop of subdomain 2 of mammalian actin is involved in the contacts important for polymerization, a chimeric *Pf*Act1 with a mammalian D-loop had a 4-fold-lower critical concentration, probably due to the higher assembly and lower disassembly rates. Like *Pf*Act1, the central residues in the *Tv*Actin D-loop were divergent from those in mammalian or yeast actins (see Fig. S1 in the supplemental material), probably explaining the high critical concentration of *in vitro* r*Tv*Actin polymerization. Moreover, *Tv*Actin and *Pf*Act1 have similar assembly rate constants (*Pf*Act1, 3.8 $\pm$ 1 subunits/$\mu$M $\cdot$ s; *Tv*Actin, 2.87 $\pm$ 0.6 subunits/$\mu$M $\cdot$ s), revealing their potential common actin dynamics (44). In contrast, *Acanthamoeba* actin with a D-loop similar to that of mammalian actin had a lower critical concentration (45), implicating the determinant of the D-loop in actin dynamics. Here, we show that r*Tv*Actin enables the formation of actin filaments that are tens of microns long, sufficient to span the whole trophozoites of both the flagellate and amoeboid forms of *T. vaginalis*, but they might be maintained in a more complicated way *in vivo*.

Compared to the nonadherent T1 isolate, more *Tv*FACP$\alpha$ and $\alpha$-actin were detected in adherent TH17 isolates, but less *Tv*FACP$\alpha$ cosedimented with F-actin (Fig. S9), possibly explaining why the adherent isolate displayed more active cytoskeleton behaviors than the nonadherent isolate. Furthermore, the adherent isolate may require a larger *Tv*FACP$\alpha$ reservoir to immediately modulate cytoskeleton dynamics in response to sudden environmental challenges.

One of the known functions of the perinuclear actin cap is to govern nuclear location and movement during nuclear division (38). When there was colocalization of *Tv*FACP$\alpha$ and F-actin at the leading edge of the extending pseudopodia, with a Pearson value of 0.82 $\pm$ 0.03, there was less colocalization observed near the actin cap, with a Pearson value of 0.32 $\pm$ 0.09 (Fig. 5B), suggesting the distinct regulations of *Tv*FACP$\alpha$ in the peripheral motile structure and the central juxtanuclear actin cap.

The human CKIP-1 protein containing a pleckstrin homology domain directs CP$\alpha$ to the cell membrane periphery and bridges the interaction of CP$\alpha$ with CKII kinase to coregulate cell morphology (34, 35). *Tv*FACP$\alpha$ Ser2, identified as a CKII phosphorylation site, is conserved with Ser9 on human or yeast CP$\alpha$ (Fig. S3D) (34, 35, 46). Human CP$\alpha$ Ser9 has been demonstrated to be phosphorylated by CKII kinase but does not directly affect actin assembly (34), indicating the divergent regulation of human CP$\alpha$ and *Tv*FACP$\alpha$. Also, yeast CP$\alpha$ Ser9 resides in the stalk domain but not the actin-binding domain; thus, Ser2 phosphorylation may not directly interfere with *Tv*FACP$\alpha$ actin binding, instead altering function by an allosteric effect or by binding with other interacting partners (46).

F-actin assembly was repressed in the hypophosphorylation mimic S2A mutant but was restored to nearly the basal level instead of exceeding it in the hyperphosphorylation mimic S2D mutant. This implies the presence of additional pathways promoting F-actin assembly. The *Tv*Fim1 protein reveals a function opposite that of *Tv*FACP$\alpha$ in accelerating F-actin polymerization, which favors phagocytosis and migration in *T. vaginalis* (36). Intriguingly, lower *Tv*FACP(pS2) levels were detected in the nonadherent T1 isolate than in the adherent TH17 isolate (Fig. S15A). However, S2D overexpression did not significantly change F-actin assembly (Fig. S15B), amoeboid morphogenesis (Fig.

S15C), and cytoadherence in the T1 isolate (Fig. S15D), strongly supporting the existence of different pathways evoking actin assembly. Two CP$\alpha$-homologous proteins (TVAG_470230 and TVAG_212270) with 32% sequence similarity (Fig. S4) and five CP$\beta$ proteins with 40% sequence similarity were identified in the database, but whether they are functionally redundant remains to be investigated. The higher eukaryotic CP was heterodimerized from the $\alpha$- and $\beta$-subunits, in which both C-terminal ~30 amino acids were organized into a mobile extension structure referred to as the tentacle, which is crucial for binding the F-actin barbed end (29, 30). A C-terminal deletion and mutation to form CP with either single tentacle can act in a way in which $\alpha$ is more important than $\beta$. In analyses of slime mold and chicken CP functions, synthetic peptides or GST fusion proteins with sequences corresponding to the $\alpha$- or $\beta$-tentacle alone inhibited 2 $\mu$M actin polymerization at a micromolar level (29, 30), supporting our *in vitro* observation that His-*Tv*FACP$\alpha$ alone may be sufficient to inhibit *Tv*Actin polymerization. A recent model demonstrated that *Plasmodium berghei* CP$\alpha$ (*Pb*CP$\alpha$) forms an atypical homodimer with a partially redundant activity to the heterodimer (*Pb*CP$\alpha\beta$) that may rescue F-actin capping in life cycle stages, while CP$\beta$ is downregulated (47). Both homo- and heterodimeric *Pb*CP genes regulate actin dynamics without changing the critical concentration of malaria actin, which is similar to our observations. The differences in the mechanism of *Tv*FACP$\alpha$ regulation requires further investigation. The *in vitro* polymerization assay revealed that *Tv*FACP$\alpha$ alone represses ~80% of actin polymerization at a higher molar ratio than that of the *Dictyostelium discoideum* CP$\alpha\beta$ heterodimer (48), possibly explaining why the overexpression of *Tv*FACP$\alpha$ alone only partially inhibits *Tv*Actin polymerization and cytoskeleton behaviors of *T. vaginalis*.

The opportunistic amoeba *Naegleria fowleri* exists in three life stages: flagellate trophozoite, amoeba trophozoite, and cyst. The growth temperature, cation level, steroid hormones, or chemical agents affect flagellate-to-amoeba transformation (49–51). In *T. vaginalis*, other than the contact-dependent effect (14, 19, 36), the factors that trigger the morphological transition are virtually unknown. Although they have the cognate behavior of flagellate-amoeba conversion, their regulation in these two protozoa may be distinct. The immediate conversion to motility may allow the parasite to rapidly respond to environmental fluctuations or flushing by humoral fluid flow in the urogenital tract (40).

Mass spectrometry data revealed that GAPDH is a major interacting partner of *Tv*Actin. In chicken neuron cells, GAPDH acts as a chaperone for $\alpha$-actin and cotranslocates with $\alpha$-actin to specialized axon sites for polymerization (52). In yeast, GAPDH associates with $\alpha$-actin and the RpB7 subunit of RNA polymerase II to regulate transcription (53, 54). The significance of GAPDH complexed with the actin cytoskeleton in *T. vaginalis* remains to be studied.

**Conclusion.** In conclusion, *Tv*FACP$\alpha$ may directly bind G- or F-actin to block actin filament extension (Fig. 10), with Ser2 phosphorylation on *Tv*FACP$\alpha$ decreasing actin-binding activity and triggering actin polymerization. In adherent *T. vaginalis* trophozoites, *Tv*FACP$\alpha$ spatially colocalizes with actin molecules at the membrane periphery of motile protrusive pseudopodia, where *Tv*FACP$\alpha$ regulates actin assembly dynamics to control the cytoskeleton behaviors of motility switching, amoeboid migration, or cytoadherence consequent to morphogenesis. The Ser2 phosphorylation status is crucial for the function of *Tv*FACP$\alpha$ in the regulation of cytoskeleton behaviors. Cytoskeleton-driven activities are also inhibited by a cytoskeleton (LatB) or CKII (TBB) inhibitor. These findings may provide potential therapeutic targets for cytoskeleton aspects to prevent *T. vaginalis* colonization and transmission.

## MATERIALS AND METHODS

**Cell cultures.** *T. vaginalis* trophozoites were cultured in Trypticase-Yeast Extract-Iron-Serum (TYI) medium at 37°C (55). Two *T. vaginalis* isolates, nonadherent isolate T1 (55) and adherent isolate TH17, were used in this study. T1, with only flagellate trophozoites, swims freely in a medium suspension. TH17 displayed vigorous morphogenesis and tightly adhered to the glass surface of a culture tube. Once the void surface was saturated by adhered trophozoites, the unbound parasites in the flagellate form swam freely in the medium suspension (Fig. 1; see also Videos S1 and S2 in the supplemental material). The flagellate trophozoites in the medium suspension and adherent trophozoites on the culture tube surface were collected for analysis as described below. The clinical isolate was obtained from a vaginitis patient.

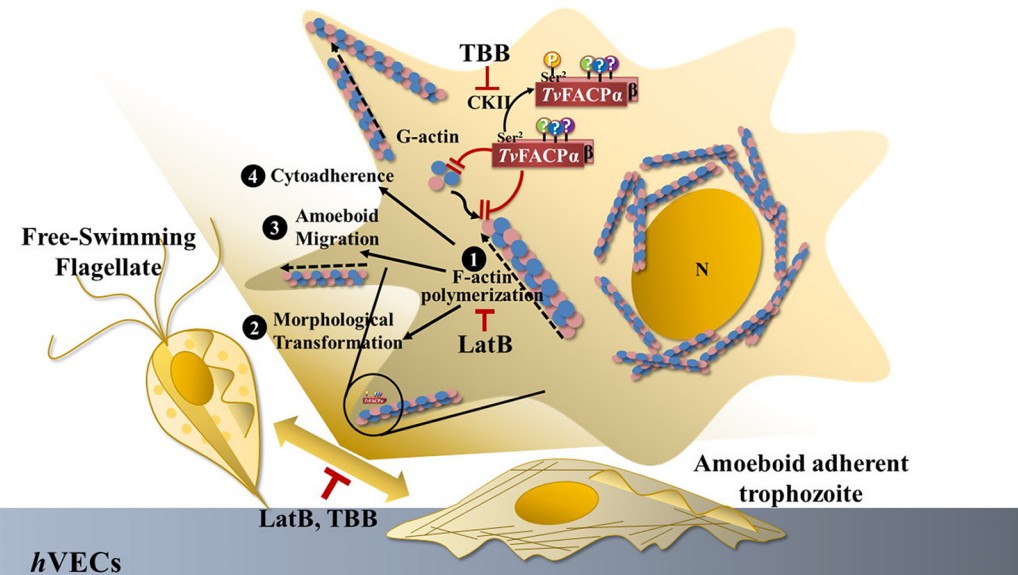

**FIG 10** Proposed model for *Tv*FACPα function and regulation. *Tv*FACPα is an actin-binding protein containing a C-terminal actin-binding domain and CKII-dependent Ser2 phosphorylation. *Tv*FACPα interacts directly with G-actin and F-actin through the actin-binding domain, and Ser2 phosphorylation is the essential signal triggering the dissociation of *Tv*FACPα and α-actin. *Tv*FACPα colocalizes with actin at the leading edge of the peripheral motile protrusions, inhibiting actin filament polymerization (1), leading to the diminishment of flagellate-amoeboid transformation and motility switching (2), amoeboid migration (3), and cytoadherence (4) in this parasite. As expected, the above-mentioned behaviors were also inhibited by TBB and LatB, supporting the significance of CKII and cytoskeleton activities for parasitism. Tight adherence and immediate migration conversion may be approaches adopted by this parasite to counteract environmental fluctuations or evade host defenses. This novel mechanism of *T. vaginalis* cytoadherence may provide new therapeutic targets for future treatment.

The vaginal swab was inoculated into TYI medium supplemented with 1,000 U penicillin, 1,000 $\mu$g/mL streptomycin, 1,000 $\mu$g/mL kanamycin, and 2.5 $\mu$g/mL amphotericin B. In contrast to the T1, T016 (56), and TH17 experimental long-term-cultured strains, this undomesticated fresh isolate (FC) was cultured short term for weeks before analysis. Human vaginal epithelium cells (hVECs) (VK2/E6E7) (ATCC CRL-2616; American Type Culture Collection) were authenticated by short tandem repeat (STR) profiling and were *Mycoplasma* negative. hVECs were cultivated in keratinocyte-serum-free medium (KSFM; Thermo Fisher Scientific, MA, USA) at 37°C in 5% $CO_2$. hVECs and all parasite lines were examined periodically for *Mycoplasma* contamination using a commercial mycoplasma PCR detection kit (Sigma-Aldrich, MA, USA).

**Preparation of lysates from adherent amoeboid and nonadherent flagellate trophozoites.** Approximately $2 \times 10^7$ trophozoites from the adherent TH17 isolate were inoculated into a culture tube with 15 mL of medium and incubated at 37°C for 2 h. The free trophozoites in the suspension were transferred to a new tube and recovered by centrifugation. The cell pellet was lysed in 1 mL lysis buffer (1% Triton X-100, 1× protease inhibitor cocktail, 1× phosphatase inhibitor cocktail, 100 $\mu$g mL$^{-1}$ TLCK [N$\alpha$-p-tosyl-L-lysine chloromethyl ketone], and 5 mM EDTA in Tris-buffered saline [TBS]). The trophozoites adhering to the glass tube were directly lysed by adding 1 mL lysis buffer and vigorously vortexing the mixture for 5 min at 4°C.

**Plasmid construction.** The full-length coding sequence of the *Tvfacpα* gene (TVAG_470230) was amplified from *T. vaginalis* genomic DNA using the primer pair *Tv*FACPα-BamHI-5′ and *Tv*FACPα-XhoI-3′ (Table 2). The PCR product was gel purified, digested with BamHI/XhoI, and ligated into the BamHI/XhoI-predigested Flp-HA-*Tv*CyP2 or pET28a backbone plasmid to obtain the Flp-HA-*Tv*FACPα or pET28-His-*Tv*FACPα plasmid. Using a similar procedure, the DNA fragments were amplified from Flp-HA-*Tv*FACPα

**TABLE 2** Oligonucleotide primers used in this study[a]

| Primer | Sequence |
|---|---|
| *Tv*FACPα-BamHI-5′ | 5′-AA<u>GGATCC</u>ATGAGCGAGAGCGAAAAT-3′ |
| *Tv*FACPα-XhoI-3′ | 5′-AA<u>CTCGAG</u>TTAGCACTTCATGCCACC-3′ |
| *Tv*FACPαΔ237–261-3′ | 5′-AA<u>CTCGAG</u>ACGAAGCTGGAAAAGAAC-3′ |
| *Tv*FACPαS2A-5′ | 5′-AA<u>GGATCC</u>ATGgccGAGAGCGAAAATATC-3′ |
| *Tv*FACPαS2D-5′ | 5′-AA<u>GGATCC</u>ATGgatGAGAGCGAAAATAT-3′ |
| *Tv*Actin-BamHI-5′ | 5′-AA<u>GGATCC</u>ATGGCTGAAGAAGACGTTCAGAC-3′ |
| *Tv*Actin-XhoI-3′ | 5′-AA<u>CTCGAG</u>TTAGAAGCACTTGCGGTGGAC-3′ |
| *Tv*Cadherin-BamHI-5′ | 5′-<u>GGATCC</u>ATGATTTGGACTTTTTTATTGCAG-3′ |
| *Tv*Cadherin-XhoI-3′ | 5′-<u>CTCGAG</u>TTACTTTCTAAGCCAAAGAATTATTACT-3′ |

[a]The restriction sites are underlined, and the mutation sites are indicated in lowercase type.

individually using the primer pairs *Tv*FACPαS2A-5′ and *Tv*FACPα-XhoI-3′ for the S2A mutation, *Tv*FACPαS2D-5′ and *Tv*FACPα-XhoI-3′ for the S2D mutation, and *Tv*FACPα-BamHI-5′ and *Tv*FACPαΔ237–261-3′ for the actin-binding domain deletion mutant (Δ237–261). The PCR products were gel purified and subcloned into the Flp-HA-*Tv*FACPα or pET28a backbone with BamHI/XhoI sites to generate the Flp-HA-*Tv*FACPα(S2A), Flp-HA-*Tv*FACPα(S2D), Flp-HA-*Tv*FACPα(Δ237–261), or pET28-His-*Tv*FACPα(Δ237–261) plasmid.

To express HA-tagged α-actin in *T. vaginalis* or glutathione *S*-transferase (GST)-fused α-actin for the GST pulldown or actin polymerization assays, the full-length coding sequence of the *Tv*actin gene (TVAG_337240) was amplified from *T. vaginalis* genomic DNA by using the primer pair *Tv*Actin-BamHI-5′ and *Tv*Actin-XhoI-3′. The gel-purified PCR product was digested with BamHI and XhoI and then ligated into the BamHI- and XhoI-predigested Flp-HA-*Tv*FACPα or pGST-*Tv*CyP2 plasmid (57) to generate the Flp-HA-*Tv*Actin or pGST-*Tv*Actin plasmid.

The *Tv*Cadherin expression plasmid was constructed, and the coding sequence of the *Tv*cadherin gene (TVAG_393390) (2) was amplified from *T. vaginalis* genomic DNA by using the primer pair *Tv*Cadherin-BamHI-5′ and *Tv*Cadherin-XhoI-3′ and subcloned into the Flp-HA-*Tv*FACPα backbone vector with BamHI and XhoI sites to produce the Flp-HA-*Tv*Cadherin plasmid.

**Cytoadherence binding assay.** hVECs were cultured in a 24-well plate to an 85% confluent monolayer. Mid-log-phase *T. vaginalis* trophozoites prelabeled with 5 $\mu$M carboxyfluorescein diacetate succinimidyl ester (CFSE) dye (CellTrace; Thermo Fisher Scientific, MA, USA) were suspended in 200 $\mu$L KSFM and inoculated into the hVEC culture at a multiplicity of infection (MOI) of 2:1. At the specified time points, the medium was aspirated, and unbound trophozoites were removed by washing two times with phosphate-buffered saline (PBS) for 5 min each. CFSE-labeled parasites were detected by using an inverted fluorescence microscope equipped with a mercury lamp and a color filter for CFSE (LD Achroplan 20×/0.4 PH2 objective and Axiovert 200M microscope; Zeiss, Oberkochen, Germany). The cell morphology was recorded in the phase-contrast mode, and the fluorescence intensity was measured by using ImageJ (version 1.53q; National Institutes of Health, MD, USA). The fluorescence from the cytoadherence assay before the removal of unbound trophozoites was detected as the input control, and the relative intensities of bound versus input parasites from five random microscopic fields were averaged to quantify cytoadherence.

**Real-time microscopy.** The activity of trophozoites cocultured with the hVEC monolayer in a glass-bottom culture dish was monitored in real time by inverted microscopy in the phase-contrast mode with a sampling rate of 1 frame per 15 s over time, as indicated (LD Achroplan 40×/0.5 PH2 objective and Axiovert 200M microscope; Zeiss, Oberkochen, Germany).

**Inhibitor treatment.** LatB (1 $\mu$M) (Sigma-Aldrich, MA, USA) or TBB (0.58 $\mu$M) (Sigma-Aldrich, MA, USA) was added to the *T. vaginalis* culture, and the culture was incubated at 37°C for 2 h before analysis. The viabilities of the parasites with or without drug treatment were evaluated by a trypan blue exclusion assay. The trophozoites were stained with 0.4% trypan blue in PBS. The percentage of viable cells was calculated from 500 trophozoites from five independent microscopic fields.

**Morphology analysis.** Trophozoites were cultured on a glass slide in a humid chamber at 37°C for 1 h, and the morphology was observed by phase-contrast microscopy (LCAch N 20×/0.40 PHP ∞/1/FN22 objective and CKX31 microscope; Olympus, Tokyo, Japan). The percentage of flagellate or amoeboid forms was measured from 600 trophozoites within 12 random microscopic fields. Compared to flagellate trophozoites, which have a solid spherical shape and a diameter of <10 $\mu$m, trophozoites with a stretching diameter of >10 $\mu$m and a morphology transforming into a distinctive irregular appearance or a flat round-disk form lying on the glass surface are defined as the amoeboid form of *T. vaginalis* trophozoites.

**Immunofluorescence assay.** *T. vaginalis* trophozoites were fixed with 4% formaldehyde and permeabilized with 0.2% Triton X-100. The samples were then incubated with the primary antibodies rabbit anti-α-actin (1:200) (GenScript, NJ, USA), mouse anti-α-actin (1:400) (clone Ac-40; Abcam, Cambridge, UK), mouse anti-HA (1:200) (clone HA-7; Sigma-Aldrich, MA, USA), mouse anti-AP65 (1:400) (7), rabbit anti-PFO (1:400) (10), and rabbit anti-*Tv*FACPα (1:400), followed by reactions with fluorescein isothiocyanate (FITC)- or Cy3-conjugated goat anti-mouse or -rabbit IgG secondary antibodies (1:200) (Jackson ImmunoResearch, PA, USA). The specimens were air dried and mounted in medium with 4′,6-diamidino-2-phenylindole (DAPI) (Vector Laboratories, CA, USA) for observation by confocal microscopy (Plan-Apochromat 100×/1.40 oil Ph3 objective and LSM-700 microscope; Zeiss, Oberkochen, Germany). The fluorescence signal was detected with an argon-ion laser (excitation/emission wavelengths [Ex/Em] of 488/520 nm for FITC, 555/640 nm for Cy3, and 405/460 nm for DAPI), and the image was captured in a single z-slice.

**F-actin staining.** Trophozoites were fixed with 4% formaldehyde and then permeabilized with 0.2% Triton X-100. The sample was incubated with 20 $\mu$g mL$^{-1}$ of TRITC-conjugated phalloidin (Sigma-Aldrich, MA, USA) diluted in PBS with 1% bovine serum albumin (BSA) in the dark at room temperature for 1 h. After washing three times with PBS, the glass slide was air dried and mounted in antifade medium (Vector Laboratories, CA, USA) for fluorescence microscopy with a mercury lamp (UPlanFl 100×/1.3 oil Iris objective and BX-60 microscope; Olympus, Tokyo, Japan).

**Signal colocalization evaluation.** The distribution of the fluorescence intensity from the immunofluorescence assay (IFA) was analyzed by a plot profile in ImageJ (version 1.53q; National Institutes of Health, MD, USA). The plot was generated and Pearson's correlation coefficient was calculated using Microsoft Office Excel 2019 software, with values of 1 indicating perfect colocalization, −1 indicating anticorrelation, and 0 representing no correlation.

**Western blotting.** The protein samples denatured in $1\times$ sodium dodecyl sulfate (SDS) sample buffer were separated by SDS-polyacrylamide gel electrophoresis (PAGE) in a 12% gel before being blotted onto a polyvinylidene difluoride (PVDF) membrane using a wet transblot system (Bio-Rad, CA, USA). The blocked membrane was incubated with the primary antibodies mouse anti-HA (1:2,000) (clone HA-7; Sigma-Aldrich, MA, USA), mouse anti-$\alpha$-actin (1:20,000) (clone Ac-40; Abcam, Cambridge, UK), mouse anti-*Tv*CyP2 (1:5000) (57), mouse anti-$\alpha$-tubulin (1:10,000) (clone DM-1A; Sigma-Aldrich, MA, USA), rabbit anti-*Tv*FACP$\alpha$ (1:3,000), mouse anti-$6\times$His (1:2,000) (clone AD1.1.10; Abcam, Cambridge, UK), rabbit anti-phospho-CKII substrate [(pS/pT)DXE] (1:1,000) (Cell Signaling Technology, MA, USA), rabbit anti-PFO (1:5,000) (10), mouse anti-AP65 (1:10,000) (7), mouse anti-GAPDH (1:10,000) (58), and mouse anti-$\alpha$-actinin (1:5,000) (58) at 4°C overnight, followed by HRP-conjugated anti-mouse or -rabbit IgG secondary antibodies (1:5,000) (Jackson ImmunoResearch, PA, USA) at 37°C for 1 h. The membranes reacted with the enhanced chemiluminescence (ECL) substrate (Thermo Fisher Scientific, MA, USA) were detected and quantified by using a UVP image system (ChemiDoc-It 815 imager and VisionWorksLS 8.6 software; Analytik Jena Company, Jena, Germany).

**Protein dephosphorylation.** The sample was incubated with 1 and 10 U of calf intestine alkaline phosphatase (Sigma-Aldrich, MA, USA) in reaction buffer (100 mM NaCl, 5 mM MgCl$_2$, 100 mM Tris-HCl [pH 9.5]) at 37°C for 30 min. The sample was denatured in $1\times$ SDS sample buffer for Western blotting.

**Immunoprecipitation.** Briefly, $6 \times 10^7$ trophozoites were lysed in 1 mL of lysis buffer (1% Triton X-100, $1\times$ protease inhibitor cocktail, $1\times$ phosphatase inhibitor cocktail, 100 $\mu$g mL$^{-1}$ TLCK, and 5 mM EDTA in TBS), centrifuged to remove unbroken cell debris before the addition of 20 $\mu$L of anti-HA antibody-conjugated agarose beads (Sigma-Aldrich, MA, USA), and then incubated on a rotator at 4°C overnight. The beads were recovered by centrifugation and washed three times with 1 mL lysis buffer. The precipitates were denatured in $1\times$ SDS sample buffer for Western blotting or staining (55, 57).

**Label-free quantitative proteomic analysis.** The proteins separated by SDS-PAGE were fixed in methanol for SYPRO ruby staining (Thermo Fisher Scientific, MA, USA) and visualization by using the Typhoon9410 imaging system (GE Healthcare, IL, USA). Each gel lane was equally cut into 4 pieces and then sliced into smaller 1-mm$^3$ cubes. The gel cubes were desalted by five washes sequentially in 1 mL of 20 mM triethylammonium bicarbonate (TEABC) buffer and 1 mL of 20 mM TEABC with 50% acetonitrile, with vigorous vortexing. The samples were sequentially reduced in 20 mM dithiothreitol (DTT) at 56°C for 1 h, alkylated in 55 mM iodoacetamide in the dark at room temperature for 30 min, and digested with trypsin (Promega, WI, USA) at 37°C overnight. The tryptic peptides were extracted by vortexing three times sequentially in 20%, 50%, and 100% acetonitrile and then dried in a vacuum concentrator (SpeedVac; Thermo Fisher Scientific, MA, USA) for liquid chromatography-tandem mass spectrometry (LC-MS/MS) analysis (55). The protein abundance from the mass spectrometry data was analyzed by a label-free quantitative method by a Mascot search, which provides an automated calculation of the exponentially modified protein abundance index (emPAI) to estimate the coverage of the identified peptides and the abundance of each protein in a data set. The identified proteins with an emPAI score of >0.25 or that were specific in the copulldown sample and their functional categories are summarized in Table 1.

**In silico analysis of protein sequence and function.** The functions of the proteins identified by mass spectrometry were categorized by the Protein Analysis through Evolutionary Relationships (PANTHER) classification system (www.pantherdb.org/). The *Tv*FACP$\alpha$ protein homologs were searched in TrichDB (https://trichdb.org/trichdb/app). The multiple-protein-sequence alignment was analyzed by using Vector NTI AdvanceR 11.5.1 software (Thermo Fisher Scientific, MA, USA). The protein search was performed by using the Basic Local Alignment Search Tool (BLAST) (https://blast.ncbi.nlm.nih.gov/Blast.cgi) or UniProt (www.uniprot.org/).

**TvFACP$\alpha$ antibody production.** The recombinant full-length His-*Tv*FACP$\alpha$ protein was produced and purified according to a standard protocol as suggested by the supplier (Qiagen, Hilden, Germany) (55, 57). The use of the purified His-*Tv*FACP$\alpha$ protein for antibody production is a customized service provided by the manufacturer (Genetex, CA, USA). The antibody specificity of anti-*Tv*FACP$\alpha$ was tested by Western blotting, as shown in Fig. 5A.

**Production of recombinant TvActin.** An *E. coli* BL21 culture (2 L) (optical density at 600 nm [OD$_{600}$] = 0.6) was induced with 1 mM isopropyl-$\beta$-D-thiogalactopyranoside (IPTG) and shaken at 37°C for 3 h. The *E. coli* culture was washed with 30 mL prechilled TBS followed by centrifugation at $23,000 \times g$ at 4°C for 20 min. The bacteria were suspended in 15 mL STE buffer (10% sucrose, 100 mM Tris [pH 8.0], 1.5 mM EDTA), and lysozyme was then added to 100 $\mu$g/mL. After incubation for 15 min at 4°C, the cells were added to 135 mL lysis buffer (0.2% Sarkosyl, 15 mM triethanolamine [pH 8.0], 50 mM NaCl, 2.5 mM ATP, 1 mM GDP, 1 mM DTT, $1\times$ protease inhibitor cocktail) (43). After stirring for 5 min, the lysate was briefly sonicated, and insoluble proteins were removed by centrifugation at $23,000 \times g$ for 20 min at 4°C. A final concentration of 2% octyl glucoside was added to the supernatant, and MgCl$_2$ and CaCl$_2$ were added to final concentrations of 1.25 mM and 1.06 mM, respectively. After stirring for 30 min at 4°C, the sample was centrifuged at $60,000 \times g$ for 10 h at 4°C. The supernatant was collected and dialyzed into G-buffer (0.1 mM CaCl$_2$, 0.2 mM ATP, 0.5 mM DTT, 2 mM Tris-HCl [pH 8.0]) with five changes of 800 mL G-buffer for 6 h each. The precipitated proteins and polymerized actin were removed from the dialyzed sample by ultracentrifugation at $100,000 \times g$ for 1 h. The supernatant was incubated with 1 mL glutathione-conjugated Sepharose beads (GE Healthcare, IL, USA) with gentle rotation at 4°C overnight. After washing twice with 3 mL low-salt (LS) wash buffer (0.1% Triton X-100, 20 mM NaCl, 0.1 mM CaCl$_2$, 2 mM Tris-HCl [pH 8.0]) and twice with 3 mL high-salt (HS) wash buffer (0.1% Triton X-100, 400 mM NaCl, 0.1 mM CaCl$_2$, 2 mM Tris-HCl [pH 8.0]), GST-*Tv*Actin was eluted in G-buffer containing 10 mM glutathione for *in vitro* assays.

The purity of the GST-*Tv*Actin-Sepharose beads was examined by using an SDS-PAGE gel stained

with Coomassie blue, and GST-*Tv*Actin was then adjusted to ~1 mg/mL in thrombin cleavage buffer (2 mM CaCl$_2$, 50 mM Tris-HCl [pH 8.0]) and reacted with thrombin (Abcam, Cambridge, UK) at a ratio of the enzyme to the target protein of 1:100 (wt/wt) at room temperature for 10 h. After the removal of GST-Sepharose beads by centrifugation at 1,000 × *g* for 10 min, the supernatant was passed through a 1-mL heparin-Sepharose-packed column (Abcam, Cambridge, UK) two times to remove thrombin from the cleavage reaction mixture. The flowthrough was dialyzed into G-buffer and then gel filtrated in a Sephacryl S-300 HR column (Sigma-Aldrich, MA, USA) with G-buffer. The protein abundance was monitored as the $A_{290}$, and the fractions from the second half of the peak absorbance to those down to 10% were collected to obtain monomeric G-r*Tv*Actin and concentrated with a 3-kDa-cutoff Vivaspin column for further assays.

**Fluorescent dye labeling of r*Tv*Actin and His-*Tv*FACPα.** r*Tv*Actin in G-buffer (~1 mg/mL) was added to a 1/10 volume of 10× F-buffer (500 mM KCl, 20 mM MgCl$_2$, 10 mM ATP, 100 mM Tris [pH 7.5]) to polymerize at room temperature for 40 min. The F-actin solution was dialyzed into labeling buffer (50 mM KCl, 1 mM MgCl$_2$, 1 mM EGTA, 0.2 mM ATP, 10 mM Tris-HCl [pH 7.0]) for 8 h. A final concentration of 100 μM *N*-(1-pyrene)iodoacetamide or Alexa Fluor 488 maleimide (Thermo Fisher Scientific, MA, USA) was added to the r*Tv*Actin solution, and the mixture was incubated with gentle rotation in the dark at 4°C overnight. The labeling reaction was quenched by the addition of 10 mM DTT, and the precipitated dye was then removed by centrifugation at 2,000 × *g* for 20 min. The supernatant was dialyzed into G-buffer for 48 h with at least three buffer exchanges and then concentrated using a 3-kDa-cutoff Vivaspin column (GE Healthcare, IL, USA) (59, 60). The labeled r*Tv*Actin solution was centrifuged at 100,000 × *g* for 1 h at 4°C to remove the remaining actin filaments (60). Also, His-*Tv*FACPα (1 mg/mL) diluted in labeling buffer (50 mM KCl, 1 mM MgCl$_2$, 1 mM EGTA, 0.2 mM ATP, 10 mM Tris-HCl [pH 7.0]) was reacted with 300 μM Alexa Fluor 555 maleimide (Thermo Fisher Scientific, MA, USA) with stirring in the dark at 4°C overnight and then quenched with 10 mM DTT. The insoluble dye was removed by centrifugation at 2,000 × *g* for 20 min, dialyzed in labeling buffer to remove the unreacted dye, and then concentrated using a 3-kDa-cutoff Vivaspin column (GE Healthcare, IL, USA).

***In vitro* polymerization of pyrene-labeled r*Tv*Actin.** Ninety microliters of a G-r*Tv*Actin solution (10% pyrene labeled) was placed into a 96-well black microplate (BD, NJ, USA), 10 μL of 10× F-buffer (500 mM KCl, 20 mM MgCl$_2$, 10 mM ATP, 100 mM Tris [pH 7.5]) was added, and the fluorescence was immediately detected by using a fluorescence spectrophotometer (SpectraMax i3x; Molecular Devices, CA, USA) at an Ex/Em of 365/410 nm in real time with a detection rate of 10 s for 50 min (60).

**TIRF microscopy.** Glass coverslips were sonicated in 60°C double-distilled water (ddH$_2$O) for 45 min and then sequentially incubated in 1 M KOH at 40°C for 3 h and 1 M HCl at 40°C overnight with slow agitation. After being thoroughly rinsed with ddH$_2$O, the coverslips were sonicated in 70% ethanol followed by 96% ethanol. The coverslips were stored in 100% ethanol and air dried before use. To assemble the flow cell, the coverslip was attached to a glass slide raised by two small strips of parafilm spaced ~10 mm apart. Next, 50 μM myosin II (Cytoskeleton Inc., CO, USA) was inactivated with 1 mM *N*-ethylmaleimide (NEM) at room temperature for 1 h and at 4°C overnight. The reaction was quenched with 10 mM DTT at 4°C for 1 h. The flow cell was coated with 0.25 μM NEM-inactivated myosin II in HS-TBS (50 mM Tris-HCl [pH 7.6], 600 mM NaCl) for 2 min, followed by 1% BSA in HS buffer for 2 min and 1% BSA in LS buffer (50 mM Tris-HCl [pH 7.6], 50 mM NaCl) for 2 min, and rinsed with 1× TIRF buffer (10 mM imidazole [pH 7.0], 50 mM KCl, 1 mM MgCl$_2$, 1 mM EGTA, 1 mM ATP, 20 mM DTT, 15 mM glucose, 20 μg/mL catalase, 100 μg/mL glucose oxidase, and 0.25% methylcellulose [4,000 cP]). To exchange Ca-ATP-actin for Mg-ATP-actin, a 1/10 volume of 10× ME buffer (1 mM MgCl$_2$, 2 mM EGTA) was added to the G-r*Tv*Actin (10% Alexa Fluor 488-labeled) solution. After 2 min, the sample was mixed with an equal volume of 2× TIRF buffer, immediately introduced into the flow cell, and placed under a microscope (Plan-Apochromat 100/1.4 oil differential interference contrast [DIC] VIS objective and Carl Zeiss laser TIRF 3 system; Zeiss, Oberkochen, Germany) for imaging with a DPSS laser at a capture rate of 15 to 20 s per frame over time (60, 61). Kymographs were generated from representative elongating filaments by using ImageJ software to calculate the actin assembly rate (1-μm actin filament = ~370 actin subunits). A plot of the actin assembly rate versus the concentration was generated from 25 growing filaments from each of two sample preparations, with the slope indicating the assembly rate constant, the *y* intercept indicating the disassembly rate constant, and the *x* intercept indicating the critical concentration (44).

**Negative staining and TEM.** Four micromolar r*Tv*Actin polymerized in 1× F-buffer for 10 min or 1 h was 50-fold diluted, and a 20-μL aliquot was then absorbed onto a carbon-coated copper grid for 5 min. After washing with F-buffer twice and H$_2$O twice, the grid was negatively stained with 1% uranyl acetate for 1 min at room temperature (62). The sample was observed by TEM (H7500; Hitachi, Tokyo, Japan) at a ×20,000 or a ×40,000 magnification at 100 kV. The image was captured with an AMT camera system.

**G-actin-binding assay.** Briefly, 4 μM GST or gel-filtrated monomeric GST-*Tv*Actin immobilized on 20 μL of glutathione-conjugated Sepharose 4B beads (GE Healthcare, IL, USA) was incubated with 80 nM His-*Tv*FACPα or His-Δ237–261 in 1 mL of G-buffer with 0.2% Triton X-100 at 4°C with rotation overnight. The GST beads were washed three times with 1 mL of G-buffer with 0.2% Triton X-100 and then denatured in 1× SDS sample buffer for Western blotting (63). Alternatively, 100 μL of 0.8 μM monomeric r*Tv*Actin or BSA in G-buffer was added to a 96-well microplate and incubated with gentle shaking at 4°C for 8 h. After three washes with PBS-Tween (PBST) (0.05% Tween 20 in PBS), the samples were blocked in PBST with 5% nonfat milk at 37°C for 1 h. Next, 100 μL of 0.4 μM His-*Tv*FACPα or His-Δ237–261 was added to the well, and the mixture was incubated at 4°C with gentle shaking overnight. The unbound protein was removed by three washes with PBST, and the plate was incubated with anti-6×His primary antibody (2,000×) (clone AD1.1.10; Abcam, Cambridge, UK) and HRP-conjugated goat anti-mouse IgG secondary antibody at room temperature for 2 h. The well was washed three times with PBST, and 100 μL/well of 3,3′,5,5′-tetramethylbenzidine (TMB) substrate (Sigma-Aldrich, MA, USA) was added at

room temperature for 5 min. The colorimetric reaction was stopped with 100 $\mu$L/well of 1 N HCl, and the $OD_{450}$ was determined by spectrophotometry (SpectraMax i3x; Molecular Devices, CA, USA).

**F-actin cosedimentation assay.** A 1/10 volume of 10× F-buffer (500 mM KCl, 20 mM MgCl$_2$, 10 mM ATP, 100 mM Tris [pH 7.5]) was added to 6 $\mu$M G-*Tv*Actin in 50 $\mu$L G-buffer at room temperature for 1 h to complete actin polymerization. F-r*Tv*Actin was diluted in 1 mL 1× F-buffer and reacted with 80 nM His-*Tv*FACP$\alpha$ or His-$\Delta$237–261 with gentle rotation at 4°C overnight. The samples were sedimented by ultracentrifugation at 100,000 × $g$ (62). F-r*Tv*Actin in the pellet and G-r*Tv*Actin in the supernatant were detected by Western blotting.

**Actin biochemical fractionation.** G-actin and F-actin were fractionated and enriched using a commercial *in vivo* assay Biochem kit (Cytoskeleton Inc., CO, USA), according to the manufacturer's instructions, with minor modifications. Briefly, ~3 × 10$^7$ trophozoites were incubated in cell lysis buffer (Cytoskeleton Inc., CO, USA) with vigorous agitation at 4°C for 30 min and homogenized by a 23-gauge needle on a 5-mL syringe. The total lysate was centrifuged at 1,000 × $g$ to remove the unbroken cell debris, followed by ultracentrifugation at 100,000 × $g$ for 1 h to separate the insoluble F-actin and associated proteins in the pellet (P) from the soluble G-actin in the supernatant (S). For Western blotting, $\alpha$-tubulin and *Tv*CyP2 were detected as purity markers for the S and P fractions, respectively. The intensities of $\alpha$-actin in the S and P fractions were first normalized to the relative intensities of their *Tv*CyP2 and $\alpha$-tubulin signals, respectively. The ratio of the $\alpha$-actin signal intensity of the individual fraction (S or P) to that of the combination (S plus P) was calculated to evaluate the G/F-actin content in the parasite.

**Cell migration assay.** For the wound-healing assay, adherent *T. vaginalis* trophozoites were cultured to a confluent monolayer in a T25 flask. A scratch (200 $\mu$m to 1 mm wide) was generated by scraping the trophozoite monolayer with a P200 tip. After the removal of cell debris by washing once with growth medium, the culture flask was incubated at 37°C, and images were captured in a defined area at an interval of 30 min over 2 h. The wound closure area in each image was measured by using ImageJ software (version 1.53q; National Institutes of Health, MD, USA). For the transwell migration assay, ~1 × 10$^7$ trophozoites suspended in 2 mL of TYI medium were inoculated into the top insert divided by a polyester membrane with 3-$\mu$m pores (4.6 cm$^2$; JET Biofil, Guangzhou, China). The top insert was placed into a 6-well culture plate containing 2 mL of TYI medium and cultured at 37°C for 30 min. The trophozoites in the top insert and bottom well were collected for microscopic observation and Western blotting.

**Statistical analysis.** The statistical significance of the data collected from control and conditional samples was analyzed by using Microsoft Office Excel 2019 software with Student's *t* test. A *P* value of <0.05 is considered a significant difference.

**Data availability.** All data generated or analyzed during this study are included in the manuscript and supplemental material. The mass spectrometry proteomics raw data have been deposited to Dryad (https://datadryad.org/stash/share/e30mZQElM-nBNmJOniuiGSBJWBkB7V4-t0XzQ891cX8).

## SUPPLEMENTAL MATERIAL

Supplemental material is available online only.

**SUPPLEMENTAL FILE 1**, DOCX file, 4.9 MB.
**SUPPLEMENTAL FILE 2**, MP4 file, 2.8 MB.
**SUPPLEMENTAL FILE 3**, MP4 file, 3.1 MB.
**SUPPLEMENTAL FILE 4**, MP4 file, 13.1 MB.

## ACKNOWLEDGMENTS

We are grateful to Jung-Hsiang Tai (Institute of Biomedical Sciences, Academia Sinica, Taiwan) for the *T. vaginalis* T1 isolate; John Alderete (Washington State University, USA) for the anti-$\alpha$-actinin, anti-GAPDH, and anti-AP65 antibodies; and Rossana Arroyo (CINVESTA, Mexico City, Mexico) for the anti-PFO antibody. Also, we are grateful to the Proteomics Core Facility (Institute of Biomedical Sciences, Academia Sinica, Taiwan) for the LC-MS/MS analysis, Shao-Chun Hsu (Imaging Core Facility at the First Core Labs, College of Medicine, National Taiwan University) for the technical support on TIRF microscopy, and Horng-Tzer Shy (Department of Anatomy and Cell Biology, College of Medicine, National Taiwan University) for the technical support on electron microscopy.

Kai-Hsuan Wang, Investigation, Validation, and Methodology. Jing-Yang Chang, Investigation, Validation, and Methodology. Fu-An Li, Investigation, Validation, and Methodology. Kuan-Yi Wu, Investigation, Validation, and Methodology. Shu-Hao Hsu, Investigation and Methodology. Yen-Ju Chen, Investigation, Validation, and Methodology. Tse-Ling Chu, Investigation and Validation. Jessica Lin, Investigation and Validation. Hong-Ming Hsu, Investigation, Validation, Project Administration, Supervision, Funding Acquisition, Conceptualization, Writing – Original Draft Preparation, and Writing – Review & Editing.

This work was supported by grants from the National Science and Technology Council of Taiwan (110-2320-B-002-048 and 110-2320-B-002-076) and National Taiwan University (NTUJP-112L7226).

We declare that we have no competing interests in the manuscript.

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
