## [Reviewer comments · Microbiology Spectrum]

Microbiology Spectrum

An atypical F-actin capping protein modulates cytoskeleton behaviors crucial to *Trichomonas vaginalis* colonization

Kai-Hsuan Wang, Jing-Yang Chang, Fu-An Li, Kuan-Yi Wu, Shu-Hao Hsu, Yen-Ju Chen, Tse-Ling Chu, Jessica Lin, and Hong-Ming HSU

Corresponding Author(s): Hong-Ming HSU, Department of Tropical Medicine and Parasitology, College of Medicine, National Taiwan University

Review Timeline:

Submission Date:	February 9, 2023
Editorial Decision:	March 22, 2023
Revision Received:	May 20, 2023
Accepted:	May 22, 2023

Editor: Björn Kafsack

Reviewer(s): Disclosure of reviewer identity is with reference to reviewer comments included in decision letter(s). The following individuals involved in review of your submission have agreed to reveal their identity: Naomi Morrissette (Reviewer #1)

Transaction Report:

DOI: <https://doi.org/10.1128/spectrum.00596-23>

March 22, 2023

Dr. Hong-Ming HSU
Department of Tropical Medicine and Parasitology, College of Medicine, National Taiwan University
Department of Tropical Medicine and Parasitology
No.1 Jen Ai road section 1
Taipei
Taiwan

Re: Spectrum00596-23 (An atypical F-actin capping protein modulates cytoskeleton behaviors crucial to *Trichomonas vaginalis* colonization)

Dear Dr. Hong-Ming HSU:

Link Not Available

Sincerely,

Björn Kafsack

Journals Department
Reviewer comments:

Reviewer #1 (Comments for the Author):

This MS is a data dense characterization of capping protein in *T. vaginalis* which shows that it plays key roles in morphological transitions and cytoadherence. The data and conclusions are exciting but many of the key findings get lost in the density and display of the data. Specifically, the figures contain panels that do not work well together because the labels, size and formatting are inefficient with text that is too small or large and patches of unused blank space due to the special placement of panels. While the authors do not need to do additional experiments, they should think carefully about what data is included so that the conclusions are evident to readers.

The authors must recraft the figures to optimize viewing. All text should be the same size and font, the images and graphs should be easily visible without having to zoom in and out. The color of labels should be considered in terms of contrast. Also, some figure panels (such as Figure 1C) are not adjusted to optimize contrast and are difficult to see.

An example of how the figures could be revised is suggested here for figures 1 and 2. This is just a suggestion, but the concepts may help the authors. The impact of this work is lessened if it is hard to see and interpret the data.

Figure 1:

Panel A: by showing both CFSE and the merge with the phase image of the host cells, the 6 panels are displayed at a small size, along with the quantification. I suggest just showing the merged panels and putting an inset I or B for input or binding so that the size of the images can be increased. The scale bar does not have to include the text of its size - this can go in the legend. The height of the quantifying graph should correspond to the height of the image panels and if the text reads percent bound, it will fit on one line as the Y axis label.

Panel B scale bar size can be put in the legend and omitted from the figure to improve legibility. Again, the height of the quantifying graph should correspond to the height of the image panels. The Y axis label should be simplified to fit in a legible font size. How about percent amoeba.

Panel C should be adjusted to improve contrast and the labels for time can simply be numbers with the legend describing the timing. Again, the scale bar size can go in the legend.

Figure 2:

The graphs in panels A and C are not legible at the same scale as the blots. The labeling of the blots could be streamlined so that the graphs could be displayed at a larger size. One way to simplify everything is to make panels A and C one merged panel. This means that the labels only get shown once. The panels should be enlarged to occupy the full size of the figure - maximize your information! The quantification of colocalization (plot profile analysis) shown in D is illegible, particularly the inset. Consider if the graph should be a supplemental figure. Quantification in panel E is nearly illegible. Again, parts of panels (and all panels) should be presented at a size that can be viewed legibly.

Other feedback:

I suggest making Figure 4 a supplemental figure.

Again, for all cytoadherence assays, I would show the merge only so that the panels can be larger.

Reviewer #2 (Comments for the Author):

The authors of this paper identified a role for a parasite protein, TvFACP α , in parasite colonization via its effects on the parasite's cytoskeleton. This was accomplished by comparing a more adherent strain of the parasite with a less adherent one in a number of assays that looked at parasite morphology, adherence, motility and finally comparative proteomics. They then go on to characterize the ability of TvFACP α on actin assembly in an in vitro assay and in the parasite using both the WT and a mutated version that lacks a putative actin binding domain. Finally, the authors demonstrate that TvFACP α activity is regulated by CKII phosphorylation using chemical inhibitors as well as hypo- and hyperphosphorylation mutants of TvFACP α . The experimental data in this paper is largely described well and experimentally rigorous but could use some minor revisions to strengthen their argument.

Strengths of the Paper:

Experiments are largely well described and thorough.

Comparison a more adherent and less adherent strain of the parasites is a clever way to identify novel parasite colonization factors.

Multiple transgenic parasites are used which strengthen the argument and model the authors propose.

Major comments and concerns:

Figure 1. In panel B the authors are measuring amoeboid forms of the parasite. The criteria by which parasites are scored as amoeboid is not clear in the methods and materials section. While most readers can probably appreciate the difference by eye (see figure B images) it is still unclear how an edge case would be assessed. If this is a purely qualitative assessment it should be stated as such in the methods. If there is a quantitative assessment to define amoeboid status (such as area or roundness) this should be stated in the methods.

Figure 2. See comment for figure 1 regarding how amoeboid vs flagellate forms are determined.

Figure 3. While it may be unlikely that the inhibitor, LatB, is killing or adversely affecting the parasites given the short time frame (2-hour pretreatment) the authors should still demonstrate that LatB is not cytotoxic to the parasites as dead or dying parasite could affect the outcome of the results. The results would be strengthened with a parasite viability assay to ensure any cytotoxic effects LatB might have on the parasite are not affecting the outcome of the results in figure 3.

Figure 7. See comment for figure 1 regarding how amoeboid vs flagellate forms are determined. Like figure 3 the authors should test to make sure the inhibitor TBB is not killing the parasite which could affect the interpretation of their results.

What do the levels of TvFACP(pS2) look like in the less adherent strain of the parasite by Western Blot?

Figure 8. See comment for figure 1 regarding how amoeboid vs flagellate forms are determined. Similar to figure 3 the authors should test to make sure the inhibitor TBB is not killing the parasite which could affect the interpretation of their results.

Figure 9. Similar to figure 3 the authors should test to make sure the inhibitor TBB is not killing the parasite which could affect the interpretation of their results. Additionally, for the transwell experiment (C-E) the results would be strengthened by direct counting of the numbers of parasites that have migrated from the insert to the bottom well as western blotting is an indirect measure.

Minor comments and suggestions:

Figure 5. Labels for panel B showing concentration are difficult to read as they over lap with the data.

Figure 6-7. What would happen if you over expressed TvFACP α S2D mutant in the less adherent strain (T1)? Would it result in an increase in adherence/amoeboid morphology? This would strengthen the argument that TvFACP α plays a role in switching between amoeboid and flagellate forms.

Staff Comments:

Preparing Revision Guidelines

Please return the manuscript within 60 days; if you cannot complete the modification within this time period, please contact me. If you do not wish to modify the manuscript and prefer to submit it to another journal, please notify me of your decision immediately so that the manuscript may be formally withdrawn from consideration by Microbiology Spectrum.

Kai-Hsuan Wang, Jing-Yang Chang¹, Fu-An Li, Kuan-Yi Wu, Shu-Hao Hsu, Yen-Ju Chen, Tse-Ling Chu, Jessica Lin, and Hong-Ming Hsu, "An atypical F-actin capping protein modulates cytoskeleton behaviors crucial to *Trichomonas vaginalis* colonization"

Summary: This manuscript investigates the role of TvFACP α , the *Trichomonas* homolog of the actin binding protein CAPZ α (capping protein) in *T. vaginalis* cytoadherence, which is critical to pathogenesis. Actin, TvFACP α and adhesion were characterized in adherent and non-adherent isolates of *T. vaginalis*. Upon co-culture of *T. vaginalis* with a vaginal epithelial cell monolayer, trophozoites of the adherent isolate but not those of the nonadherent isolate convert from a flagellated form into an amoeboid form within 10 minutes. The amoeboid form moves by a crawling motility and latrunculin treatment reduces cytoadherence. These data suggest a role for actin in adhesion and amoeboid motility. However, since α -actin and α -actinin protein levels do not change between the flagellated and amoeboid forms within the adherent isolate, it is likely that changes in cytoskeletal organization are modulated by post translational modifications. Actin associated proteins were identified by co-immunoprecipitation and mass spectrometry. One of the hits was TvFACP α . In vitro biochemical assays indicate that TvFACP α inhibits pyrene actin polymerization, like CPs in metazoan organisms. However, TvFACP α has an atypical G-actin binding activity, which reduces actin assembly by sequestering subunits independent of barbed end binding/filament extension. Parasite based assays employed quantification of pelleted (polymerized) actin under conditions of over-expression and/or treatment with the CKII inhibitor TBB to investigate the predicted CKII phosphorylation site at TvFACP α S2. Overexpression of wild type or S2A (non-phosphorylated) TvFACP α reduces pelleted (F) actin but increases the co-sedimentation (association) of TvFACP α with F-actin. Expression of a phosphomimetic S2D TvFACP α does not change pelleted actin relative to the untransfected line but does show lower co-sedimentation (association). These and other experiments are consistent with the conclusion that CKII-dependent Ser2 phosphorylation causes TvFACP α and α -actin to dissociate. Both release of actin subunits and uncapping of the barbed ends of F-actin stimulate actin polymerization. Overexpression of wild type or S2A (non-phosphorylated) TvFACP α or TBB treatment reduces amoeboid morphogenesis, suggesting a role for this protein in the transition to a cytoadherent form.

Overall feedback:

This MS is a data dense characterization of capping protein in *T. vaginalis* which shows that it plays key roles in morphological transitions and cytoadherence. The data and conclusions are exciting but many of the key findings get lost in the density and display of the data. Specifically, the figures contain panels that do not work well together because the labels, size and formatting are inefficient with text that is too small or large and patches of unused blank space due to the special placement of panels. While the authors do not need to do additional experiments, they should think carefully about what data is included so that the conclusions are evident to readers.

The authors must recraft the figures to optimize viewing. All text should be the same size and font, the images and graphs should be easily visible without having to zoom in and out. The color of labels should be considered in terms of contrast. Also, some figure panels (such as Figure 1C) are not adjusted to optimize contrast and are difficult to see.

An example of how the figures could be revised is suggested here for figures 1 and 2. This is just a suggestion, but the concepts may help the authors. The impact of this work is lessened if it is hard to see and interpret the data.

Figure 1:

Panel A: by showing both CFSE and the merge with the phase image of the host cells, the 6 panels are displayed at a small size, along with the quantification. I suggest just showing the merged panels and putting an inset I or B for input or binding so that the size of the images can be increased. The scale bar does not have to include the text of its size – this can go in the legend. The height of the quantifying graph should correspond to the height of the image panels and if the text reads percent bound, it will fit on one line as the Y axis label.

Panel B scale bar size can be put in the legend and omitted from the figure to improve legibility. Again, the height of the quantifying graph should correspond to the height of the image panels. The Y axis label should be simplified to fit in a legible font size. How about percent amoeba.

Panel C should be adjusted to improve contrast and the labels for time can simply be numbers with the legend describing the timing. Again, the scale bar size can go in the legend.

Figure 2:

The graphs in panels A and C are not legible at the same scale as the blots. The labeling of the blots could be streamlined so that the graphs could be displayed at a larger size. One way to simplify everything is to make panels A and C one merged panel. This means that the labels only get shown once. The panels should be enlarged to occupy the full size of the figure – maximize your information! The quantification of colocalization (plot profile analysis) shown in D is illegible, particularly the inset. Consider if the graph should be a supplemental figure. Quantification in panel E is nearly illegible. Again, parts of panels (and all panels) should be presented at a size that can be viewed legibly.

Other feedback:

I suggest making Figure 4 a supplemental figure.

Again, for all cytoadherence assays, I would show the merge only so that the panels can be larger.

The authors of this paper identified a role for a parasite protein, *TvFACP* α , in parasite colonization via its effects on the parasite's cytoskeleton. This was accomplished by comparing a more adherent strain of the parasite with a less adherent one in a number of assays that looked at parasite morphology, adherence, motility and finally comparative proteomics. They then go on to characterize the ability of *TvFACP* α on actin assembly in an *in vitro* assay and in the parasite using both the WT and a mutated version that lacks a putative actin binding domain. Finally, the authors demonstrate that *TvFACP* α activity is regulated by CKII phosphorylation through the use of chemical inhibitors as well as hypo- and hyperphosphorylation mutants of *TvFACP* α . The experimental data in this paper is largely described well and experimentally rigorous, but could use some minor revisions to strengthen their argument. However, the discussion section is very difficult to read and disjointed.

Strengths of the Paper:

Experiments are largely well described and thorough.

Comparison a more adherent and less adherent strain of the parasites is a clever way to identify novel parasite colonization factors

Multiple transgenic parasites are used which strengthen the argument and model the authors propose.

Major comments and concerns:

Figure 1. In panel B the authors are measuring amoeboid forms of the parasite. The criteria by which parasites are scored as amoeboid is not clear in the methods and materials section. While most readers can probably appreciate the difference by eye (see figure B images) it is still unclear how an edge case would be assessed. If this is a purely qualitative assessment it should be stated as such in the methods. If there is a quantitative assessment to define amoeboid status (such as area or roundness) this should be stated in the methods.

Figure 2. See comment for figure 1 regarding how amoeboid vs flagellate forms are determined.

Figure 3. While it may be unlikely that the inhibitor, LatB, is killing or adversely affecting the parasites given the short time frame (2 hour pretreatment) the authors should still demonstrate that LatB is not cytotoxic to the parasites as dead or dying parasite could affect the outcome of the results. The results would be strengthened with a parasite viability assay to ensure any cytotoxic effects LatB might have on the parasite are not affecting the outcome of the results in figure 3.

Figure 7. See comment for figure 1 regarding how amoeboid vs flagellate forms are determined. Similar to figure 3 the authors should test to make sure the inhibitor TBB is not killing the parasite which could affect the interpretation of their results.

What do the levels of *TvFACP*(pS2) look like in the less adherent strain of the parasite by Western Blot?

Figure 8. See comment for figure 1 regarding how amoeboid vs flagellate forms are determined. Similar to figure 3 the authors should test to make sure the inhibitor TBB is not killing the parasite which could affect the interpretation of their results.

Figure 9. Similar to figure 3 the authors should test to make sure the inhibitor TBB is not killing the parasite which could affect the interpretation of their results. Additionally, for the transwell experiment (C-E) the results would be strengthened by direct counting of the numbers of parasites that have migrated from the insert to the bottom well as western blotting is an indirect measure.

Minor comments and suggestions:

Figure 5. Labels for panel B showing concentration are difficult to read.

Figure 6-7. What would happen if you over expressed *TvFACP* α S2D mutant in the less adherent strain (T1)? Would it result in an increase in adherence/amoeboid morphology? This would strengthen the argument that *TvFACP* α plays a role in switching between amoeboid and flagellate forms.

Dear reviewers:

We are grateful for the valuable comments to make this manuscript more readable and convincing. The manuscript has been modified as per the reviewers' suggestions, and the related comments are responded to point by point as described below. Due to the line number shifting after PDF conversion by the submission system, you may track the issues in our attached Marked-Up manuscript PDF file.

Reviewer #1 (Comments for the Author):

This MS is a data dense characterization of capping protein in *T. vaginalis* which shows that it plays key roles in morphological transitions and cytoadherence. The data and conclusions are exciting but many of the key findings get lost in the density and display of the data. Specifically, the figures contain panels that do not work well together because the labels, size and formatting are inefficient with text that is too small or large and patches of unused blank space due to the special placement of panels. While the authors do not need to do additional experiments, they should think carefully about what data is included so that the conclusions are evident to readers.

The authors must recraft the figures to optimize viewing. All text should be the same size and font, the images and graphs should be easily visible without having to zoom in and out. The color of labels should be considered in terms of contrast. Also, some figure panels (such as Figure 1C) are not adjusted to optimize contrast and are difficult to see.

Re: We are grateful for the valuable comments. In this revised manuscript, we rearranged the figure position and optimized label size and font for better readability. Also, the image contrast of Figure 1C is adjusted to be clearly visible.

An example of how the figures could be revised is suggested here for figures 1 and 2. This is just a suggestion, but the concepts may help the authors. The impact of this work is lessened if it is hard to see and interpret the data.

Figure 1:

Panel A: by showing both CFSE and the merge with the phase image of the host cells, the 6 panels are displayed at a small size, along with the quantification. I suggest just showing the merged panels and putting an inset I or B for input or binding so that the size of the images can be increased. The scale bar does not have to include the text of its size - this can go in the legend. The height of the quantifying graph should correspond to the height of the image panels and if the text reads percent bound, it will fit on one line as the Y axis label.

Re: Cytoadherence assay only shows the merged images, magnified for better readability. The size texts are removed from the image scale bars and stated in the figure legends. Also, we adjust the

height of quantifying graphs corresponding to that of the image panel and simplify the label text on the Y axis.

Panel B scale bar size can be put in the legend and omitted from the figure to improve legibility. Again, the height of the quantifying graph should correspond to the height of the image panels. The Y axis label should be simplified to fit in a legible font size. How about percent amoeba.

Re: The size text is removed from the image scale bar and described in the figure legend. The quantifying graph is adjusted to the height corresponding to the image panel. The Y-axis label is simplified with a legible font size.

Panel C should be adjusted to improve contrast and the labels for time can simply be numbers with the legend describing the timing. Again, the scale bar size can go in the legend.

Re: The image contrast of panel C is adjusted to be clearly visible (Figure 1C).

Figure 2:

The graphs in panels A and C are not legible at the same scale as the blots. The labeling of the blots could be streamlined so that the graphs could be displayed at a larger size. One way to simplify everything is to make panels A and C one merged panel. This means that the labels only get shown once. The panels should be enlarged to occupy the full size of the figure - maximize your information! The quantification of colocalization (plot profile analysis) shown in D is illegible, particularly the inset. Consider if the graph should be a supplemental figure. Quantification in panel E is nearly illegible. Again, parts of panels (and all panels) should be presented at a size that can be viewed legibly.

Re: In Figure 2, the panel A, C, and E labels are enlarged to a legible font size. The original inset graph with the colocalization Pearson value in panel D is moved to a new panel.

Other feedback:

I suggest making Figure 4 a supplemental figure.

Re: The original Figure 4 has been moved into Supplementary Figure 3.

Again, for all cytoadherence assays, I would show the merge only so that the panels can be larger.

Re: The merged images were used for all cytoadherence assays of this revised manuscript (Figures 1A, 3D, 8C, and 8D).

Reviewer #2 (Comments for the Author):

The authors of this paper identified a role for a parasite protein, TvFACP α , in parasite colonization

via its effects on the parasite's cytoskeleton. This was accomplished by comparing a more adherent strain of the parasite with a less adherent one in a number of assays that looked at parasite morphology, adherence, motility and finally comparative proteomics. They then go on to characterize the ability of TvFACP α on actin assembly in an in vitro assay and in the parasite using both the WT and a mutated version that lacks a putative actin binding domain. Finally, the authors demonstrate that TvFACP α activity is regulated by CKII phosphorylation using chemical inhibitors as well as hypo- and hyperphosphorylation mutants of TvFACP α . The experimental data in this paper is largely described well and experimentally rigorous but could use some minor revisions to strengthen their argument.

Strengths of the Paper:

Experiments are largely well described and thorough.

Comparison a more adherent and less adherent strain of the parasites is a clever way to identify novel parasite colonization factors.

Multiple transgenic parasites are used which strengthen the argument and model the authors propose.

Major comments and concerns:

Figure 1. In panel B the authors are measuring amoeboid forms of the parasite. The criteria by which parasites are scored as amoeboid is not clear in the methods and materials section. While most readers can probably appreciate the difference by eye (see figure B images) it is still unclear how an edge case would be assessed. If this is a purely qualitative assessment it should be stated as such in the methods. If there is a quantitative assessment to define amoeboid status (such as area or roundness) this should be stated in the methods.

Re: Compared to the flagellate trophozoites with solid spherical round shape and diameter under 10 μm , the trophozoites with a stretching diameter over 10 μm and the morphology transforming into distinctive irregular appearance or flat round-disk form laying on the glass surface are defined as the amoeboid form of *T. vaginalis* trophozoites. The related description is stated in the revised Materials and Methods section (line 543 of Marked-Up Manuscript PDF file).

Figure 2. See comment for figure 1 regarding how amoeboid vs flagellate forms are determined.

Re: The flagellate/amoeboid morphogenesis is defined in the Materials and Methods section of the revised manuscript (line 543 of Marked-Up Manuscript PDF file).

Figure 3. While it may be unlikely that the inhibitor, LatB, is killing or adversely affecting the parasites given the short time frame (2-hour pretreatment) the authors should still demonstrate that

LatB is not cytotoxic to the parasites as dead or dying parasite could affect the outcome of the results. The results would be strengthened with a parasite viability assay to ensure any cytotoxic effects LatB might have on the parasite are not affecting the outcome of the results in figure 3.

Re: The cell viability assay was performed to demonstrate the little effects of LatB or TBB on the parasite's vitality. The procedure was stated in the revised Materials and Methods section (line 535), the data were shown in Supplementary Figures 2A and 11C, and the results were described in lines 159 and 278 of Marked-Up Manuscript PDF file.

Figure 7. See comment for figure 1 regarding how amoeboid vs flagellate forms are determined. Like figure 3 the authors should test to make sure the inhibitor TBB is not killing the parasite which could affect the interpretation of their results.

Re: The definition for trophozoite at amoeboid versus flagellate forms was stated in the Materials and Methods section (line 543 of Marked-Up Manuscript PDF file).

The cell viability assay was tested to demonstrate the little effect of LatB or TBB on the parasite's viability. The procedure was described in the revised Materials and Methods section (line 535 of Marked-Up Manuscript.docx), the data were shown in Supplementary Figures 2A and 11C, and the results were written in lines 159 and 278 of Marked-Up Manuscript PDF file.

What do the levels of TvFACP(pS2) look like in the less adherent strain of the parasite by Western Blot?

Re: The *TvFACP(pS2)* in the less-adherent T1 isolate was detected to a level lower than the TH17 adherent isolates. The relevant information was discussed in line 393 of Marked-Up Manuscript PDF file.

Figure 8. See comment for figure 1 regarding how amoeboid vs flagellate forms are determined. Similar to figure 3 the authors should test to make sure the inhibitor TBB is not killing the parasite which could affect the interpretation of their results.

Re: The definition for trophozoite at amoeboid versus flagellate forms was stated in the Materials and Methods section (line 543 of Marked-Up Manuscript PDF file).

The cell viability assay was tested to demonstrate the little effect of LatB or TBB on the parasite's viability. The procedure was described in the revised Materials and Methods section (line 535 of Marked-Up Manuscript.docx), the data were shown in Supplementary Figures 2A and 11C, and the results were written in lines 159 and 278 of Marked-Up Manuscript PDF file.

Figure 9. Similar to figure 3 the authors should test to make sure the inhibitor TBB is not killing the

parasite which could affect the interpretation of their results.

Re: The cell viability assay demonstrated that TBB is not killing the parasite under our test condition, as described in line 278 of Marked-Up Manuscript PDF file.

Additionally, for the transwell experiment (C-E) the results would be strengthened by direct counting of the numbers of parasites that have migrated from the insert to the bottom well as western blotting is an indirect measure.

Re: The cell count ratio between the parasite migrating into the bottom wells and remaining in the inserts was calculated to evaluate the transwell migration efficiency. The data was supplied in Supplementary Figures 14B and 14C, and the results were described in line 331 of Marked-Up Manuscript PDF file.

Minor comments and suggestions:

Figure 5. Labels for panel B showing concentration are difficult to read as they over lap with the data.

Re: The original Figure 5 was moved to Figure 4 of the revised manuscript and modified as per the reviewer's comment.

Figure 6-7. What would happen if you over expressed TvFACP α S2D mutant in the less adherent strain (T1)? Would it result in an increase in adherence/amoeboid morphology? This would strengthen the argument that TvFACP α plays a role in switching between amoeboid and flagellate forms.

Re: The F-actin content, amoeboid transition, and cytoadherence were barely affected in the T1 isolate overexpressing S2D mutant, suggesting the involvement of the different pathways in evoking actin assembly and derived actin-based cytoskeleton behaviors. TvFACP α may be a regulator essential but not sufficient for mediating actin polymerization dynamics. The data was shown in Supplementary Figure 15, and the result was discussed in line 393 of Marked-Up Manuscript PDF file.

Staff Comments:

Preparing Revision Guidelines

To submit your modified manuscript, log onto the eJP submission site

at <https://spectrum.msubmit.net/cgi-bin/main.plex>. Go to Author Tasks and click the appropriate manuscript title to begin the revision process. The information that you entered when you first submitted the paper will be displayed. Please update the information as necessary. Here are a few examples of required updates that authors must address:

Please return the manuscript within 60 days; if you cannot complete the modification within this time period, please contact me. If you do not wish to modify the manuscript and prefer to submit it to another journal, please notify me of your decision immediately so that the manuscript may be formally withdrawn from consideration by Microbiology Spectrum.

May 22, 2023

Dr. Hong-Ming HSU
Department of Tropical Medicine and Parasitology, College of Medicine, National Taiwan University
Department of Tropical Medicine and Parasitology
No.1 Jen Ai road section 1
Taipei
Taiwan

Re: Spectrum00596-23R1 (An atypical F-actin capping protein modulates cytoskeleton behaviors crucial to *Trichomonas vaginalis* colonization)

Dear Dr. Hong-Ming HSU:

Your manuscript has been accepted, and I am forwarding it to the ASM Journals Department for publication. You will be notified when your proofs are ready to be viewed.

Sincerely,

Björn Kafsack
Editor, Microbiology Spectrum
